# SKILL-MIX : A FLEXIBLE AND EXPANDABLE FAMILY OF EVALUATIONS FOR AI MODELS

**Dingli Yu**[1]   **Simran Kaur**[1]   **Arushi Gupta**[1]
**Jonah Brown-Cohen**[2]   **Anirudh Goyal**[2]   **Sanjeev Arora**[1]
[1]Princeton Language and Intelligence (PLI), Princeton University
[2]Google DeepMind

## ABSTRACT

With LLMs shifting their role from statistical modeling of language to serving as general-purpose AI agents, how should LLM evaluations change? Arguably, a key ability of an AI agent is to flexibly combine, as needed, the basic skills it has learned. The capability to combine skills plays an important role in (human) pedagogy and also in a paper on emergence phenomena (Arora & Goyal, 2023).

This work introduces SKILL-MIX , a new evaluation to measure ability to combine skills. Using a list of $N$ skills the evaluator repeatedly picks random subsets of $k$ skills and asks the LLM to produce text combining that subset of skills. Since the number of subsets grows like $N^k$, for even modest $k$ this evaluation will, with high probability, require the LLM to produce text significantly different from any text in the training set. The paper develops a methodology for (a) designing and administering such an evaluation, and (b) automatic grading (plus spot-checking by humans) of the results using GPT-4 as well as the open LLaMA-2 70B model.

Administering a version of SKILL-MIX to popular chatbots gave results that, while generally in line with prior expectations, contained surprises. Sizeable differences exist among model capabilities that are not captured by their ranking on popular LLM leaderboards ("cramming for the leaderboard"). Furthermore, simple probability calculations indicate that GPT-4's reasonable performance on $k = 5$ is suggestive of going beyond "stochastic parrot" behavior (Bender et al., 2021), i.e., it combines skills in ways that it had not seen during training.

We sketch how the methodology can lead to a SKILL-MIX based eco-system of open evaluations for AI capabilities of future models. We maintain a leaderboard of SKILL-MIX at `skill-mix.github.io`.

## 1   INTRODUCTION

As LLMs shift roles from mere statistical models of language to fairly general-purpose AI agents, the inadequacy of existing evaluations—even those introduced within the past year—has become clear. Yet, leading models routinely score over $90\%$ on many evaluations (OpenAI, 2023).

Current evaluations on leaderboards test quantitative, scientific, and academic reasoning suffer from serious limitations. Since they originate from human tests or textbooks, they are vulnerable to *training-set contamination*, i.e., examples very similar to the ones on the evaluation ending up in the training corpus of the model (OpenAI, 2023; Li, 2023). This is hard to measure given the immense size of training corpora and the fact that the corpora are rarely released. The contamination issue especially affects evaluations based upon human exams (whose difficulty is tied to being time-limited and closed-book) since models are now being trained on technical textbooks as well as course materials.[1]

A variant of the contamination issue is "*cramming for the leaderboard.*" It is possible to deliberately train a model on data similar to those used in the leaderboard evaluations. Such datasets are easy to generate from a small number of examples using existing strong models. If "cramming" happens

---

[1]Studies have revealed significant anomalies on GPT-4's performance on questions exams created after its release (Narayan & Kapoor).

| Generation | Grading |
|---|---|
| **Input:** Write at most 2 sentences in the context of sewing that illustrate all of the following skills: modus ponens, red herring, and metaphor. | **Input:** Could you please help me grade a student's response to the following question? |
| **Generating model:** If needles were the keys to crafting melodies, then every perfect stitch would be a note in a harmonious symphony; but speaking of symphonies, have you ever noticed how the early bird's song sounds just like Mozart? | Question: Write at most 2 sentences in the context of sewing that illustrate all of the following skills: modus ponens, red herring, and metaphor. |
| | Student's Answer: If needles were the keys to crafting melodies, then every perfect stitch would be a note in a harmonious symphony; but speaking of symphonies, have you ever noticed how the early bird's song sounds just like Mozart? |
| | Grading rubric: Correctly illustrates skills (3 points). On topic (1 point). Makes sense (1 point). At most 2 sentences (1 point). |
| | **Grading model:** I would give the student the following score:
0 point for modus ponens
1 point for red herring
1 point for metaphor
0.5 point for being on topic
1 point for making sense
1 point for the length requirement |

Figure 1: **Left: Simplified depiction (with simplified prompt) of the generation stage of our evaluation.** The full prompt appears in Appendix C.3.1. The generating model is given a topic (sewing) as well as skills (modus ponens, red herring, metaphor), and asked to generate text demonstrating the skills. The full prompt contains skill definitions and examples, which can be found in Appendix C.3.1. **Right: Simplified depiction (with simplified prompt) of the grading stage of our evaluation.** The grading model (not necessarily the same as the generating model) is given the generating model output and grading instructions, and returns pointwise grading. The full grading prompt can be found in Appendix C.3.2.

during pre-training, it becomes hard to detect. If it happens during fine-tuning, it may be detectable if it ends up harming general-purpose language skills.

Yet another issue arising from the secrecy of the training corpus is that it is difficult to verify how original the model's text productions truly are. In a recent interview (Hinton & Ng), Hinton suggested that a significant hurdle in current discussions of AI risk is absence of agreement among experts on whether or not models have already gone past "stochastic parrots" behavior (Bender et al., 2021)—i.e., whether they are able to actually understand the world, or at a minimum produce novel thoughts or "behavior" that they did not see in the training corpus.

**Desiderata for next-generation evaluations:** To sum up, we need evaluations that are: (a) clearly relevant to general-purpose intelligence and language understanding; (b) easy for humans to design and administer, including with academic-level resources; (c) resistant to training-set contamination and "cramming for the leaderboard;" (d) capable of revealing LLM's capability on producing novel text that they did not see in the training corpus (Hinton & Ng), i.e., going beyond "stochastic" parrots behavior (Bender et al., 2021); (e) easy to grade at scale (while allowing human spot-checking); (f) easily upgradeable into harder evaluations in the future as models get stronger; and (g) comprehensible at some level for the general public, including with respect to points (c) and (d).

## 1.1 SKILL-MIX

Our SKILL-MIX evaluation tests the model's ability to produce sensible text satisfying natural constraints. It starts with a set of $N$ skills (an example is: "use of metaphor") that every LLM could reasonably be expected to have encountered in its training set —say, because each skill has a Wikipedia entry. SKILL-MIX also uses a list of $T$ topics that have low, but non-negligible, probability in any reasonable training corpus –e.g., *"dueling," "gardening."* SKILL-MIX $(k)$ consists of randomly picking a subset of $k$ skills out of $N$, and a random topic out of $T$ topics. The chat agent is then asked to produce a short piece of text (say, 3 sentences) in the context of the selected topic and illustrate all $k$ selected skills. This evaluation can be easily administered using any set of skills and questions and any $k$. The model is being required to do highly constrained text generation, whose difficulty intuitively increases with $k$ [2]

---

[2]The authors of this paper took an average of more than 7 minutes to answer SKILL-MIX (4). See Table 15 for some examples of text generated by humans vs. GPT-4.

Since the evaluation task consists of sampling a random $k$-tuple from a distribution, performance can be estimated reliably with a few hundred runs of the model. Thus human grading is feasible with modest budgets. We found that it is just as reliable to do auto-grading followed by human spot-checking of the grading. We used two easily-available models GPT-4 (OpenAI, 2023)) and LLaMA-2-70B-Chat (Touvron et al., 2023).

SKILL-MIX is easily tailored to a model's capability by adjusting $k$, and in general larger models are able to handle larger $k$ (Section 1.2). In the future, as models get stronger, the evaluation can be made harder by not only increasing $k$ but also by adding new skills to our list, or by switching to more difficult or specialized skills, or to topics that are rarer or more specialized. It is also possible to create SKILL-MIX evaluations for special domains (e.g., coding or science) and for multi-modal data, although we leave that for future work.

SKILL-MIX has two theoretical inspirations. The first are theories of human pedagogy and learning (Forehand, 2005; Koedinger et al., 2012; 2023; Li et al., 2013), which have long focused on the ability to combine skills. The other is a recent paper (Arora & Goyal, 2023) that gave a theory for how complex skills emerge in LLMs when they are scaled up. The paper assumed that understanding small pieces of text consists of applying $k$ skills out of a large list of basic skills. Under some assumptions, the paper showed that scaling up language models leads to improvement in proficiency at applying $k'$-tuples of basic skills, where $k'$ increases with scaling. (Roughly speaking, $k'$ doubles when the mode is scaled up 10x.) Its appendix included a few examples of text produced by current chatbots in response to requests to combine a given set of skills, and that was an inspiration for our SKILL-MIX .

**Difficulty of** SKILL-MIX **with increasing** $k$**.** The intuition that the hardness of SKILL-MIX increases with $k$ can be supported by a simple calculation. Given a list of $N$ skills, there are $\binom{N}{k}$ ways to choose the subset of $k$ skills. For $N = 1000$, this quantity exceeds $10^{10}$ when $k = 4$, and $10^{12}$ when $k = 5$. Furthermore, the skills and topics are fairly rare in the corpus —e.g., the word "sushi" has a unigram probability of $10^{-7}$ in Google n-grams (Google, 2012). Thus, the chance that the training corpus contains a short piece of text on the chosen topic that exhibits the chosen set of $k$ skills becomes very small even for $N = 100$ (see Section 6). Such calculations are very relevant to be convinced that a model is going past "stochastic parrots" behavior as well as has resistance to dataset contamination.

**Paper Organization.** Section 2 describes related prior work. Section 3 lays out design principles for the evaluation: assembling the list of skills and topics, as well as the prompt used to administer it. Section 4 describes the setup for auto-grading. Section 5 reports the experimental results. Section 6 gives the calculation for verifying that the model is going beyond stochastic parrots behavior. Section 7 sketches a future framework of several SKILL-MIX evaluations maintained by disinterested research groups, with a secret list of skills and topics.

## 1.2 KEY FINDINGS

Section 5 shows results of administering SKILL-MIX ($k$) for different $k$ to today's leading models. For proprietary models, the results are mostly in accord with prior expectations. GPT-4 ranks highest, being able to perform reasonably even for $k = 5$. LLaMA-2-70B-Chat turns in a good performance, somewhat below GPT-3.5 Turbo, which is in line with general consensus about its capabilities. For most models, $k = 3$ already proves too tough.

**Evidence of cramming for the leaderboard.** Hugging Face's Open LLM leaderboard (Beeching et al., 2023), which is based upon EleutherAI's evaluation harness (Gao et al., 2021), is seen as a proving ground for open LLMs. Many models currently at the top of the leaderboard are LLaMA-2 derivatives, and are ranked much higher than the corresponding LLaMA-2 model. However, on SKILL-MIX these models perform poorly and worse than LLaMA-2-70B-Chat, suggestive of cramming that significantly harmed general-purpose text skills (see Section 5). The recent Falcon-180B-Chat (Almazrouei et al., 2023) also places higher on the leaderboard than LLaMA-2-70B-Chat, and has been claimed to have capabilities between GPT-3.5-turbo and GPT-4 based upon this ranking. Yet, it fares worse than LLaMA-2-70B-Chat on SKILL-MIX . Mistral-7B-Instruct-v0.1 also did not live up to claims of being significantly better than the corresponding LLaMA model.

**Detecting saturation of an evaluation.** AlpacaEval (Li et al., 2023) is a popular evaluation (with an accompanying leaderboard) for text generation. It uses GPT-4 and a dataset of prompts to check how often the model's generated output "wins" against that of DaVinci003. AlpacaEval presents a good case-study on the difficulties of creating good evaluations. Designed to give small open models

a fighting chance in early 2023, it shows signs of saturation a mere 6 months later. Even 13B models now win against DaVinci003 with probability exceeding 90%, and recently Xwin-LM-70B-V0.1 (built on the LLaMA-2 base models) climbed to the top, pushing GPT-4, by a hair, to second place. Did LLM capabilities truly progress this much within 6 months? We find that the new champion scores well on SKILL-MIX , and noticeably better than LLaMA-2-70B-Chat, but it is handily beaten by GPT-4. In general, most reasonable models now get similar scores on AlpacaEval, but show widely varying performances on SKILL-MIX , which suggests that AlpacaEval has lost its discriminative ability. SKILL-MIX avoids this shortcoming by evaluating on constrained text generation, and using $k$ to adjust the difficulty.

**Demonstrating capabilities beyond "stochastic parrot."** Whether or not current models are capable of going past "stochastic parrots" behavior (Bender et al., 2021) remains a crucial topic of discussion in the field (Hinton & Ng), and it is unclear what this would mean. In context of SKILL-MIX we define it to mean: *ability to correctly use combinations of skills + topic that it had not seen in the training corpus.* Our version of SKILL-MIX uses a list of skills each of which has a Wikipedia entry or listing (and thus known to every LLM). As mentioned, GPT-4 generates correct answers with reasonable probability for SKILL-MIX ($k = 5$), and we estimate that at least $1/3$ of the time this involves text that combines a mix of skills and topics that had not appeared in the training corpus. For $k = 6$, we estimate the majority of the correct answers generated by GPT-4 to involve novel combinations (see Section 6 and Appendix D for calculations). We are unable to find similar evidence for any other model at this time.

**Design principles for SKILL-MIX .** We include ablation studies suggesting design principles for new versions of SKILL-MIX : (i) auto grading setup (Section 4) (ii) deducting points for explicitly mentioning skill names in the answer (Appendix C.6) (iii) filtering out common skills (Appendix C.6). Our initial experiments involved a set of 101 skills selected from books and Wikipedia. Our calculation in Section 6 suggests that including skills that have higher frequencies in the training corpus makes SKILL-MIX easier. We identified 17 skills that appeared with rather high frequency in the common crawl corpus. Omitting them from the evaluation appeared to make it much harder for most models (see Section C.6). This second version is recommended.

## 2 PRIOR WORK

Arora & Goyal (2023) gives a mathematical model for skill emergence via LLM scaling. (In principle it applies to non-text data as well, since it makes almost no assumptions about what "text" or "skills" are.) The key assumption is that pieces of text involve random combinations of skills, and then reductions in cross-entropy (which can happen on an arbitrary subset of text pieces) are shown to imply improvements in both individual skills and combinations of skills. A key implication is that the trained model may show competence on k-tuples of skills even though this k-tuple of skills was not demonstrated in the training set. (Note that individual skills were demonstrated, as were some random k-tuples, but most k-tuples were not demonstrated.) It is important to note that the theory did not need to specify what it means to "combine" skills.

**Skill-It** The goal of Skill-It (Chen et al., 2023) is to select an optimal subset of the training data such that an LLM trained on this subset will perform well on downstream tasks. Chen et al. (2023) utilize the notion of *skill ordering* to construct this subset. Skill ordering refers to the natural notion that learning "simpler" skills first can make learning "difficult" skills later easier for the learner. Chen et al. (2023) define an *ordered skill set* as "a collection of skills with a directed skills graph that is neither complete nor empty, where an edge from a prerequisite skill to a skill exists if the amount of training it takes to learn the skill can be reduced if the prerequisite skill is also learned."

**Skills in Reinforcement Learning** Reinforcement learning also has a notion of "skills," which is distinct from the notion that we use (Arora & Goyal, 2023), but bears some similarities to the notion of skill used in Chen et al. (2023). In particular, Pertsch et al. (2021) aims to learn a *skill prior*, i.e., a distribution over skills, such that a model trained to develop these skills will later perform well on downstream tasks.

**Prior evaluations** Prior work has emerged which evaluates LLMs on particular skills, in a different sense than we do. Ruis et al. (2023) finds that LLMs do not do well on implicature.

**Pedagogy work** Another important area where skills have been previously studied is that of human skill learning in pedagogy. Koedinger et al. (2023) develops a cognitive and statistical model of skill acquisition with the goal of understanding why/if some students learn faster than others. Koedinger

et al. (2012) presents an algorithm to discover cognitive models, which are essentially skill models. Li et al. (2013) uses a computational model of student learning (a simulated student) in order to discover cognitive models (i.e., "learn the skills").

## 3 DESIGN OF SKILL-MIX

### 3.1 PICKING THE SKILLS

We obtained a set of 101 language skills and a set of 100 topics for SKILL-MIX evaluation. Since the goal of our evaluation is to test general-purpose text generation capability rather than the ability of the particular skills and topics, we only release 10 skills and 10 topics randomly sampled from the two sets to avoid potential "cramming" for SKILL-MIX . The randomly sampled skills and topics appear in Appendix A (see Tables 2 and 3).

We curated the topic list by first sampling a large list of topics (e.g. using Reddit forums as inspiration), and then narrowing the list down to 100 topics based on the unigram frequency of the topic (and all related synonyms) on Google Ngram viewer (Google, 2012). To earn a spot on our list, a topic's average unigram had to be around $10^{-6}$ —low (further reducing the likelihood that the model had seen the $k$ skills demonstrated in the context of the topic), and yet still ensuring good coverage even in 100B-sized corpora. The list of basic skills was designed to contain language skills which have a Wikipedia entry or listing (and thus known to every LLM), and whose definition the average person could understand. We started with a longer list of skills taken from textbooks on logical reasoning, rhetoric, theory of mind, (Kelly., 2014; Cummings, 2005; Nichols & Stich, 2003; Mueller, 2014). We tried to eliminate skills that either were too specialized —and thus difficult to apply in the context of the fairly narrow topics already chosen— or difficult to combine with other skills. (Some examples of discarded skills appear in the Appendix C.2.) For each skill, we created a description and an illustrative example of its usage —these were taken from either a textbook or Wikipedia, though occasionally, we modified them to make them clearer or more concise.

### 3.2 PROCEDURE

Our evaluation is roughly broken down into two parts (see Figure 2). In the first part, we conduct **generation**, where a (Student) language model is given a set of $k$ skills and a topic, and asked to generate some text demonstrating the $k$ skills in the context of the provided topic. Once the (Student) language model generates some text, it then must be **graded** by a (possibly different) grading language model (i.e., Grader). A simplified version of the prompts used in the generation and grading stages is depicted in Figure 1.

**Models used for generation**  Our dataset is designed to test general skills, and hence many language models may be used in the generation step. However, since the language model must be able to respond to prompt instructions ("generate $k$ skills"), we only pick models that have been instruction-tuned. These models include LLaMA-2-7B-Chat, LLaMA-2-13B-Chat, LLaMA-2-70B-Chat (Touvron et al., 2023), GPT-3.5-turbo, GPT-4 (OpenAI, 2023), Falcon-180B-Chat (Almazrouei et al., 2023), Xwin-LM-70B-V0.1 (Xwin-LM Team, 2023), Mistral-7B-Instruct-v0.1 (Mistral AI Team, 2023), Qwen-14B-Chat (Bai et al., 2023), Tigerbot-70B-Chat (TigerResearch, 2023). Note Xwin-LM-70B-V0.1, Mistral-7B-Instruct-v0.1, and Qwen-14B-Chat were released only a few days before the deadline, and all of them claim to perform better on benchmarks compared to state-of-the-art models of even larger size.

**Models used for grading**  We find that some language models are more suitable for grading than others. Some models have difficulty recognizing the presence of skills, even when they are demonstrated correctly. We use LLaMA-2-70B-Chat and GPT-4 after manually spot-checking grading samples and ensuring they aligned with human grading.

SKILL-MIX $(k)$ consists of picking a random topic and a random subset of $k$ skills (with their respective definitions and illustrative examples) from the list described above, and asking the (Student) model to produce a short piece of text which illustrates all the $k$ skills in the context of the topic of interest. A simplified example prompt appears in Figure 1, but our final prompt includes an additional question that asks the (Student) model to look over and possibly improve its first answer. We find the second answer can be much better than the first one with proper prompt engineering. In addition, we ask the model to separate its answer and explanation with "Answer" and "Explanation." Otherwise, the model may not separate its generated answer from its explanation, hampering the parsing of the answer. We provide more details about the generation prompts in Appendix C.3.1

The authors took the test (6 questions each with $k = 4$) to assess the feasibility and difficulty level. The average time taken to understand the prompt and type the answer was more than 7 minutes. This is not an easy test for humans!

## 4 AUTO-GRADING METHOD

For each prompt sampled from SKILL-MIX $(k)$, the (Student) model's corresponding answer is graded according to the following criteria: (1) the $k$ skills are present and used properly in the output; (2) the output is on the given topic; (3) the number of sentences in the output is within the provided limit, which we set to be $k - 1$; and (4) the output is a piece of sensible text. For any of the above subtasks, partial credit can be assigned if the answer partially satisfies the requirement.

The generations by the models were graded using GPT-4 as well as LLaMA-2-70B-Chat, and these grades were then spot-checked by the paper authors. In the trial run, we focused on tweaking the method for best results and consistency using a small set of around 20 generations. As usual, the assessments generated by the grading models (especially LLaMA-2) were somewhat sensitive to the prompt. We tweaked the grading prompt by including a summarized version of the generation prompt, providing definitions and illustrative examples of the individual $k$ skills, and requesting for the graded output to follow a particular format.

While both models are creditable graders, they, like all current LLMs, were unreliable at simple arithmetic, which is important for calculating a total score. We changed the prompt to require the grader to output separate grades for individual components (proper use of skills, good fit to the topic, and producing sensible text), which were subsequently aggregated (see Figure 3) using a separate but simple Python script[3].

To require separate grades for individual components, we asked GPT-4 to provide a rubric-table style grade, whereas for for LLaMA-2, we simply asked the model to include "Point earned: 1" if the requirement is met, and "Point earned: 0" otherwise, for each rubric item in the evaluation. More details, as well as the full grading prompt, appear in Appendix C.3.2.

**Human Grading: Better?**    With a small test we conducted with five NLP researchers, we found that human grading is noisy, and human graders might need significant training so they can agree on a grading rubric. The standard deviation between the human grading is high, and even higher than the difference between the average human grading and GPT-4 grading.[4] This also indicates that GPT-4 and LLaMA-2 graders are reasonable graders compared to humans. More details of the test can be found in Appendix B.

## 5 EXPERIMENTAL RESULTS

In this section, we test various instruction-tuned models (including the LLaMA-2 family, GPT family, Falcon-180B-Chat, Xwin-LM-70B-V0.1, Mistral-7B-Instruct-v0.1, Qwen-14B-Chat, and Tigerbot-70B-Chat) on their performance on SKILL-MIX for various $k$. For convenience, we use *saturation point* to denote the value of $k$ at which a model's score in SKILL-MIX drops off.

From the experimental results, we answer the following questions

- What differences arise between grading by GPT-4 vs. LLaMA-2?
- What is the effect of increasing $k$ on SKILL-MIX performance? What is the saturation point for the instruction-tuned models?
- What is the relationship between model scale and saturation point? We are particularly interested in answering this question for the family of LLaMA-2 models, since they share the same training set and methodology.

**Setup**    We evaluate various instruction-tuned models on SKILL-MIX $(k)$ with $k = 2, 3, 4$. We continue to evaluate a model with $k = 5$ (potential also $k = 6$) if it does not saturate at $k = 4$. We use both GPT-4 and LLaMA-2-70B-Chat as Grader.

---

[3]We adjusted the point awarded for meeting the number of sentences requirement based on the ground truth. While adjustments were rare, they were more common for LLaMA-2 than GPT-4.

[4]The difference between the average human grading and LLaMA-2 grading is slightly higher than the standard deviation between humans.

Table 1: **Performance of various instruction-tuned student (generating) models on** SKILL-MIX ($k$) **graded by GPT-4.** Ratio of Full Marks/Ratio of All Skills/Skill Fraction are reported for each student model at $k = 2, 3, 4$. Evaluations on $k = 5, 6$ are skipped if the Ratio of Full Marks drops below 0.05 with smaller $k$. Details on prompts can be found in Section 3. See Table 5 for additional metrics.

| Student (generator) | $k = 2$ | $k = 3$ | $k = 4$ | $k = 5$ | $k = 6$ |
|---|---|---|---|---|---|
| LLaMA-2-7B-Chat | .05/.11/.37 | .00/.01/.25 | .00/.00/.23 | -/-/- | -/-/- |
| LLaMA-2-13B-Chat | .22/.35/.45 | .05/.05/.37 | .02/.02/.45 | -/-/- | -/-/- |
| LLaMA-2-70B-Chat | .28/.28/.62 | .02/.05/.46 | .00/.01/.44 | -/-/- | -/-/- |
| GPT-3.5-turbo | .53/.60/.75 | .20/.25/.59 | .08/.13/.54 | .04/.16/.51 | -/-/- |
| GPT-4 | **.94/.97/.96** | **.68/.70/.88** | **.52/.55/.88** | **.36/.38/.86** | **.29/.29/.84** |
| Mistral-7B-Instruct-v0.1 | .05/.16/.40 | .00/.05/.36 | .00/.04/.25 | -/-/- | -/-/- |
| Qwen-14B-Chat | .16/.19/.50 | .02/.05/.39 | .01/.01/.38 | -/-/- | -/-/- |
| Xwin-LM-70B-V0.1 | .42/.58/.67 | .22/.37/.63 | .09/.17/.56 | .08/.12/.55 | -/-/- |
| Falcon-180B-Chat | .27/.33/.53 | .00/.03/.44 | .03/.07/.38 | -/-/- | -/-/- |
| Tigerbot-70B-Chat | .06/.29/.26 | .00/.10/.19 | .01/.06/.15 | -/-/- | -/-/- |

For each SKILL-MIX ($k$), we evaluate all models on 100 ($k$ skills, 1 topic) combinations with GPT-4 grading, and 30 combinations with LLaMA-2 grading.[5] We provide each specific combination of $k$ skills to the (Student) model on three instances (see Figure 2). Each of the three generated texts are also graded three times (to reduce the randomness caused by the Grader), in total creating nine grading results for each ($k$ skills, 1 topic) (see Figure 3).

**Metrics**   Each generated text can receive up to $k + 3$ points: 1 point for each correctly illustrated skill, 1 point for sticking to the topic, 1 point for coherence / making sense, and 1 point for having at most $k - 1$ sentence. Recall that we grade each generated text three times. In each round of grading, we parse each of the criteria individually from the Grader model's output. For each criterion, we then collect the majority vote among the three grading rounds. The voted points are then converted into various metrics of interest (see Appendix C.5), some of which include

- *Ratio[6] of Full Marks*: 1 if all $k + 3$ points are earned, and 0 otherwise
- *Ratio of All Skills*: 1 if $k$ points are awarded for the $k$ skills and at least 2 points are awarded for the remaining criteria, and and 0 otherwise
- *Skill Fraction*: the fraction of points awarded for the $k$ skills if all 3 points are awarded for the remaining criteria, and 0 otherwise

We then take the maximum value of the metrics among the 3 generations for a given ($k$ skill, 1 topic) combination, and average the maximum value across all the combinations.

**Differences in Grader Scores**   From Tables 1 and 4, we clearly observe that LLaMA-2 is a more generous grader than GPT-4. We also observe that when LLaMA-2 is used as the grader, it prefers generations outputted by the LLaMA-2 family. For example, across different metrics, GPT-4 generally gives a higher score to Mistral-7B-Instruct-v0.1 than to LLaMA-2-7B-Chat, but LLaMA-2 grader gives a much higher score to LLaMA-2-7B-Chat than Mistral-7B-Instruct-v0.1 for $k \geq 3$. Overall, we found via spot-checking the output that GPT-4 is a more accurate and reliable grader than LLaMA-2.

**Increasing $k$ degrades** SKILL-MIX **performance**   We observe that the ratio of full marks and the ratio of all skills can decrease dramatically when $k$ increases. With the exception of GPT-4, GPT-3.5-turbo and Xwin-LM-70B-V0.1, all models saturate on or before $k = 3$ with GPT-4 grading. Amongst the small models, LLaMA-2-7B-Chat and Mistral-7B-Instruct-v0.1 saturate at $k = 2$. Since LLaMA-2 is more generous, the saturation point is usually delayed by 1 for LLaMA-2 grading.

**Relationship between model scale and saturation point**   We find that as capacity increases on LLaMA-2, so does the saturation point. Observe that for LLaMA-2-7B-Chat, LLaMA-2-13B-Chat, LLaMA-2-70B-Chat, the saturation points (of GPT-4 grading) are $k = 2, 3$, and 3, respectively. Additionally, for any fixed $k$ and metric type, higher model capacity corresponds to a better score

---

[5]Due to computation limits, we only evaluate Falcon-180B-Chat on 30 combinations with GPT-4 grading.

[6]This is called "ratio" because the metric is later averaged over the 30 combinations, even though this metric is 0 and 1 for a single generation.

amongst the LLaMA-2 model family. However, these observations do not necessarily hold true for models from different families. For example, Falcon-180B-Chat has more model parameters than Xwin-LM-70B-V0.1, yet the saturation point of Xwin-LM-70B-V0.1 is higher than that of Falcon-180B-Chat, and Xwin-LM-70B-V0.1 also outperforms Falcon-180B-Chat across all metrics for $k = 2, 3, 4$.

**A deviation from model rankings on popular LLM leaderboards**    Recent models (i.e., Falcon-180B-Chat, Xwin-LM-70B-V0.1, Qwen-14B-Chat, Mistral-7B-Instruct-v0.1) are often introduced with their performance evaluated on AlpacaEval or Hugging Face's Open LLM Leaderboard (which contains ARC (Clark et al., 2018), HellaSwag (Zellers et al., 2019), MMLU (Hendrycks et al., 2020), and TruthfulQA (Lin et al., 2021)), along with a comparison to the LLaMA-2 and GPT families. We find that their superior performance on those evaluations (which is often misaligned with general experience) may not extend to SKILL-MIX :

- Falcon-180B-Chat and Tigerbot-70B-Chat rank higher than LLaMA-2-70B-Chat on Open LLM Leaderboard, but perform worse on SKILL-MIX for both GPT-4 and LLaMA-2 grading. Tigerbot-70B-Chat performs even worse than LLaMA-2-13B-Chat.
- Xwin-LM-70B-V0.1 takes on first place on AlpacaEval, beating GPT-4. However, Xwin-LM-70B-V0.1 is clearly worse than GPT-4 on SKILL-MIX .
- Qwen-14B-Chat outperforms LLaMA-2-70B-Chat on MMLU, HumanEval (Chen et al., 2021) and GSM8K (Cobbe et al., 2021), but performs worse than LLaMA-2-70B-Chat for $k = 2, 3, 4$ with both GPT-4 and LLaMA-2 grading.
- Mistral-7B-v0.1 outperforms LLaMA-2 13B on all benchmarks that the Mistral AI team tested. Mistral-7B-Instruct-v0.1 (the model after instruction tuning) outperforms LLaMA-2-13B-Chat on MT-Bench (Zheng et al., 2023). Yet, the situation is reversed on SKILL-MIX .

## 6    EVIDENCE FOR GOING BEYOND "STOCHASTIC PARROTS" BEHAVIOR

As mentioned in the introduction, there is interest in whether or not current models are capable of going past "stochastic parrots" behavior. Using SKILL-MIX we are able to give a guarantee: for $k = 5$ and $k = 6$, GPT-4 is generating correct answers a good fraction of time, and a reasonable portion (say more than $1/3$) of these correct answers contain combinations of skills and topics that have never jointly appeared in the training. We are unable to find similar evidence for any other model.

Our claim is verified by upperbounding the average frequency of skills $p_s$, the average frequency of topics $p_t$, and the number of sentences in the training corpus $L$. Then in the training corpus, the total number of text piece that contains $k$ of the skills and one of the topics is bounded by[7] $p_s^k p_t \binom{N}{k} T L$. If $\alpha_k$ is the model's "Ratio of Full Marks" in SKILL-MIX $(k)$, then the number of text pieces that it successfully generates with $k$ of the skills and one of the topics is at least $\alpha_k \binom{N}{k} T$. Thus, the model is "beyond stochastic parrots" if $\alpha_k > \frac{3}{2} p_s^k p_t L$, meaning more than one-third of the successful generations contain combinations not seen in the training corpus.

After filtering out skills that have a frequency of at least 5%, we estimate that with high probability, $p_s \leq 0.0144$, $p_t \leq 0.0022$ and $L \leq 5 \times 10^{10}$ on RedPajama dataset (Computer, 2023). Thus, $p_s^k p_t L \leq 0.001$ for $k = 6$, and $p_s^k p_t L \leq 0.07$ for $k = 5$. Even after filtering out common skills as in Appendix C.6, GPT-4 still has $\alpha_6 \approx 0.08$ and $\alpha_5 \approx 0.12$, which is still "beyond stochastic parrots." Please see Appendix D for more details of the calculation and estimation.

## 7    BEST PRACTICES FOR SKILL-MIX ECOSYSTEM

SKILL-MIX differs from most existing evaluations in two ways: (1) there is no dataset per se; instead, by using $N$ skills and $T$ topics, the tasks (prompts) are generated randomly on the fly from $\binom{N}{k} T$ possible combinations; and (2) for moderate $k$, the task is not easy for humans to solve, or even to grade; at a minimum, high-quality human labor is needed.

However, note that a dataset of $O(T \log T + N \log N)$ random prompts and corresponding productions already expose (with high probability) the full set of skills and topics, as well as many interesting

---

[7]This calculation assumes independence among the occurrence of skills. Verifying $k$-wise independence becomes computationally very expensive for large $k$ but we verified it for $k = 2$.

ways to combine them. Preliminary results (in Appendix E) suggest that fine-tuning on such a dataset of synthetic productions can improve scores on SKILL-MIX . Since the goal of our evaluation is to test general-purpose text generation capability rather than ability on the particular skills and topics, we propose to release only a random subset of $10\%$ of skills and topics.

But, ultimately, one needs an evaluation ecosystem. This might consist of independent research groups developing versions of SKILL-MIX (in other words, tests that are chosen randomly from a high number of potential questions) with very different sets of skills and topics, including basic skills related to science, economics, law, etc. Releasing a random subset of, say, $10\%$ of the skills and topics of a new evaluation gives the rest of the world an idea of what it tests –and how it is reasonably distinct from other existing evaluations– while retaining its difficulty. If the research groups are seen as trustworthy, this ecosystem's continuous assessment of AI capabilities might become important inputs into policy discussions in the future.

**Difficulty of grading**    An open question in the above picture is how to grade harder versions of SKILL-MIX in the future. The obvious idea is to use the current champion model for the evaluation (provided the model retains no memory of its past interactions). But, a natural question arises: whether to trust the champion's grade for itself. This relates to an interesting debate in pedagogy (Bloom et al., 1956; Anderson & Krathwohl, 2001; Forehand, 2005): *Which is harder: to ace the exam or to grade it well?* While the answer may seem obvious in quantitative or scientific fields (i.e., acing is harder), this wasn't obvious in other fields. Today it is more broadly accepted that grading is indeed easier[8], which suggests that the champion can probably grade itself (assuming the model retains no memory of past interactions), Human spot-checking seems advisable. In our experience, GPT-4's grading capabilities for a particular $k$ seem to be better than its SKILL-MIX score on the same $k$; we plan to further investigate this relationship in future work. On a related note, we found that human spot-checking becomes much easier and more accurate if the grader includes reasoning for grading decisions.

## 8    CONCLUSIONS AND TAKEAWAYS

SKILL-MIX  is an attempt to evaluate general-purpose language capabilities, specifically, a particular sort of compositional generalization. It tests the model's ability to create text on a given topic and with a given combination of well-known skills. A key idea is that the skills and the topic are chosen randomly from a big list, which itself can be expanded in many ways to give new evaluations. Such evaluations may end up requiring the (Student) model to imagine and describe situations that does not exactly match anything seen during training. Grading this evaluation is shown to be possible with GPT-4 (and the authors spot-checked the grading). Human evaluations were sought but found to have significant variance. We will try to reduce this variance in the next run with better rubrics and guidance.

Section 5 showed that the performance of proprietary models on SKILL-MIX ($k$) generally accords with popular perceptions of their quality. The results also show (in line with the theory of Arora & Goyal (2023)) that when created by competent teams, larger models achieve a higher saturation point than smaller models (e.g., the three LLaMA-2 models, which shared their training data and methodology). The disappointing performance of Falcon-180B-Chat was an exception. Several open models show signs of being over-trained for leaderboards at the expense of general-purpose language capabilities ("cramming"). Furthermore, based on our calculations, GPT-4's reasonable performance on SKILL-MIX ($k = 5$) indicates that GPT-4 surpasses "stochastic parrot" behavior (Bender et al., 2021). Of course, no single evaluation can be considered definitive. In future work, we hope to explore whether there are "cramming" shortcuts to improve performance on SKILL-MIX .

Since current leaderboards show signs of losing their discriminative power, Section 7 sketched a vision for an ecosystem of independent SKILL-MIX evaluations specializing in different sets of skills and topics (most of which would be kept secret) that could provide trusted estimates of model capabilities —both as input to public discussions of AI and as a deterrent against "cramming for leaderboards". SKILL-MIX scores track our colleagues' subjective assessments of current models.

Soon, many AI models will be multi-modal, and we hope to design a multi-modal version of SKILL-MIX for them.

---

[8]But the variance we saw among human graders on our SKILL-MIX reminds us of the difficulty of grading exams where there is no obvious best answer.

**Acknowledgements:** This work is partly supported by NSF and ONR.

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

# A  LIST OF SKILLS AND TOPICS

Here we release a random sample of 10% of the skill list and topic list. We do not release the full lists to avoid potential "cramming" for SKILL-MIX .

Table 2: 10 randomly sampled skills from the 101 skills we used in SKILL-MIX evaluation.

| Category | Skill | Definition | Example |
|---|---|---|---|
| reasoning | self serving bias | A cognitive or perceptual process that is distorted by the need to maintain and enhance one's self esteem. | "If I do well on the exam, it's because of my academic prowess and hard work. If I do poorly, it's because the course was poorly taught, and the exam was poorly proctored." |
| rhetorical | accident (fallacy) | an informal fallacy and a deductively valid but unsound argument occurring in a statistical syllogism (an argument based on a generalization) when an exception to a rule of thumb is ignored. | Cutting people with knives is a crime. Surgeons cut people with knives. Surgeons are criminals. |
| rhetorical | complex question (loaded question with implicit assumption ) | A question that is loaded with an implicit assumption. | "Why are you lying to me?" is a question that presupposes you are lying to me. Any answer you give will force you to agree you are lying. |
| rhetorical | red herring | Introducing irrelevant points to detract attention from a question. | A member of the press asks the president why they voted to expand a welfare program. The president responds, "The strength of America is the strength of its communities, and I am proud to make our communities better places." |
| literary | metaphor | a figure of speech that, for rhetorical effect, directly refers to one thing by mentioning another. | "All the world's a stage, And all the men and women merely players" is a metaphor because it's a comparison without using "like" or "as." |
| logical | spatial reasoning | The capacity to reason about the spatial relationships between objects. | The key fit into the box. Using spatial reasoning, one can deduce that the width of the key was smaller than the width of the box. |
| logical | modus ponens | A syllogism that is of the form "If P then Q. P. Hence Q." | "If today is Tuesday, then John will go to work. Today is Tuesday. Therefore, John will go to work." |

| logical | statistical syllogism | A syllogism that argues, using inductive reasoning, from a generalization true for the most part to a particular case. | "Almost all people are taller than 26 inches. Gareth is a person. Therefore, Gareth is taller than 26 inches." |
|---|---|---|---|
| theory of the mind | emotional self regulation | a complex process that involves initiating, inhibiting, or modulating one's state or behavior in a given situation. | Examples of emotional self regulation include meditating, pausing to collect oneself before speaking, and practicing stress management. |
| physical knowledge | folk physics (common knowledge physics) | The untrained human perception of basic physical phenomena. | "If I roll the pen off of the table, it will fall to the floor." |

Table 3: 10 random sampled topics from 100 topics used in SKILL-MIX evaluation.

| Topic |
|---|
| Sewing |
| Dueling |
| The Ottoman Empire |
| Triathlons |
| Beekeeping |
| Survivalism |
| Guerilla warfare |
| Gardening |
| Knots |
| Urbanism |

## B  HUMAN GRADING TEST

We will now describe a test we conducted to measure the grading quality of human graders. Our five volunteers were Ph.D. students and Postdocs working in the field of natural language processing and large language models. They were given 5 outputs generated by GPT-4 on SKILL-MIX (4). The same grading prompt for machine grading was given to human graders, asking them to give each individual point for the criteria. In this case ($k = 4$), each output could receive at most 7 points.

For each point, we computed the mean and standard deviation among the human graders and then averaged over all the points (35 in total). The average standard deviation is 0.261. On the other hand, we compare the average of human grading with GPT-4 and LLaMA-2 grading. For each point, we computed the absolute difference between the mean among the humans and machine grading. Then, we took the average over 35 points. We found that the average difference is 0.257 for GPT-4 grading and 0.268 for LLaMA-2 grading. This means if we assume the average of human grading is the ground truth, then human graders and machine graders have similar errors.

We also observe that human graders in general give lower scores for text making sense, probably because the output is usually a shortened version of the model's first attempt. In contrast, the machine graders usually award the point for "making sense" for more than 90% of the generated answers.

## C  EXPERIMENTS

### C.1  EXPERIMENTAL DETAILS

Our evaluation is roughly broken down into two parts. In the first part, we conduct **generation**, where a language model is given a set of $k$ skills and a topic and asked to generate some text demonstrating

the $k$ skills. Once some text has been generated, it then must be **graded** by a (possibly different) language model.

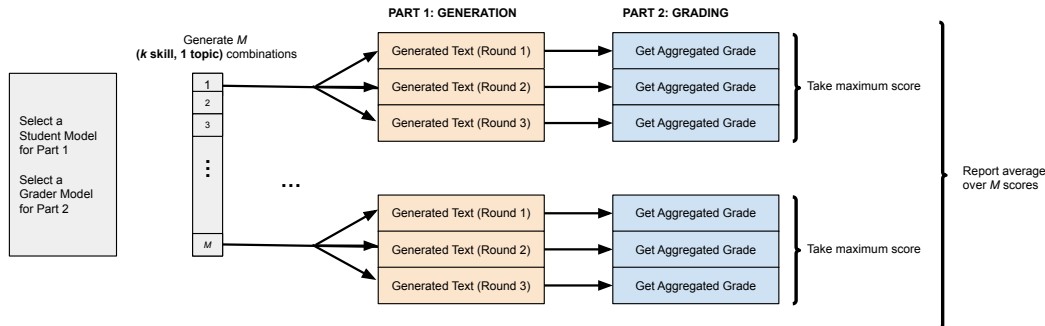

Figure 2: **Illustration of** SKILL-MIX ($k$) **pipeline.** In our experiments, we use $M = 30$. For a more detailed illustration of grading a single piece of generated text, see Figure 3.

**Models used for generation.** Our dataset is designed to test general skills, and hence many language models may be used in the generation step. However, since the language model must be able to respond to prompt instructions ("generate k skills"), we pick only models that have been instruction-tuned. These models include LLaMA-2-7B-Chat, LLaMA-2-13B-Chat, LLaMA-2-70B-Chat (Touvron et al., 2023), GPT-3.5-turbo, GPT-4 (OpenAI, 2023), Falcon-180B-Chat (Almazrouei et al., 2023), Xwin-LM-70B-V0.1 (Xwin-LM Team, 2023), Mistral-7B-Instruct-v0.1 (Mistral AI Team, 2023), and Qwen-14B-Chat (Bai et al., 2023).

**Models used for grading.** We find that some language models are more suitable for grading than others. Some models have difficulty recognizing the presence of skills, even when they are demonstrated correctly. We use LLaMA-2-70B-Chat and GPT-4 after manually spot-checking grading samples and ensuring they align with human grading.

**Details of model configurations** We do not use quantization on any of the models. For the GPT family, we use OpenAI API with default generation configuration and the minimal system prompt "`You are a helpful assistant.`" For the LLaMA-2 family, we use 2 A100 GPU and run with no system prompt, 0.7 temperature, 1.0 repetition penalty, and 512 max new tokens. For Falcon-180B-Chat, we use the prompt format mentioned in the official Huggingface blog of Falcon-180B-Chat (Schmid et al., 2023) and the same parameters as LLaMA-2 family. For Xwin-LM-70B-V0.1, we use the official prompt format (Xwin-LM Team, 2023) and the same hyperparameters as those used for the LLaMA-2 family. For Mistral-7B-Instruct-v0.1, we access the prompt format with `tokenizer.apply_chat_template` function and again the same parameters as the LLaMA-2 family. For Qwen-14B-Chat, we directly use the `model.chat` function as mentioned in their official Github repository.

## C.2 SKILL CHOOSING PROCESS

The list of $\approx 100$ skills used to draw tuples of $k$ skills was manually curated, and designed to include skills the average person could understand which were common enough that they would appear on Wikipedia (and hence in the model's pre-training data). We pick skills from standard textbooks on logic (Kelly., 2014), pragmatics (Cummings, 2005) and theory of mind (Nichols & Stich, 2003). We also pick literary skills from Wikipedia. Not all skills were considered suitable for our dataset. Because we want to evaluate the ability of models to compose skills, we eliminate skills that are trivially present in almost any piece of text. Below is an example of a skill that was eliminated from our dataset because it is so common in the English language that the model may "accidentally" generate it, thereby falsely appearing as though it is able to combine this skill with other skills.

```
Skill:  Pied Piping

Definition:  A syntax phenomenon whereby a given expression brings along an accompanying
phrase when it is moved.
```

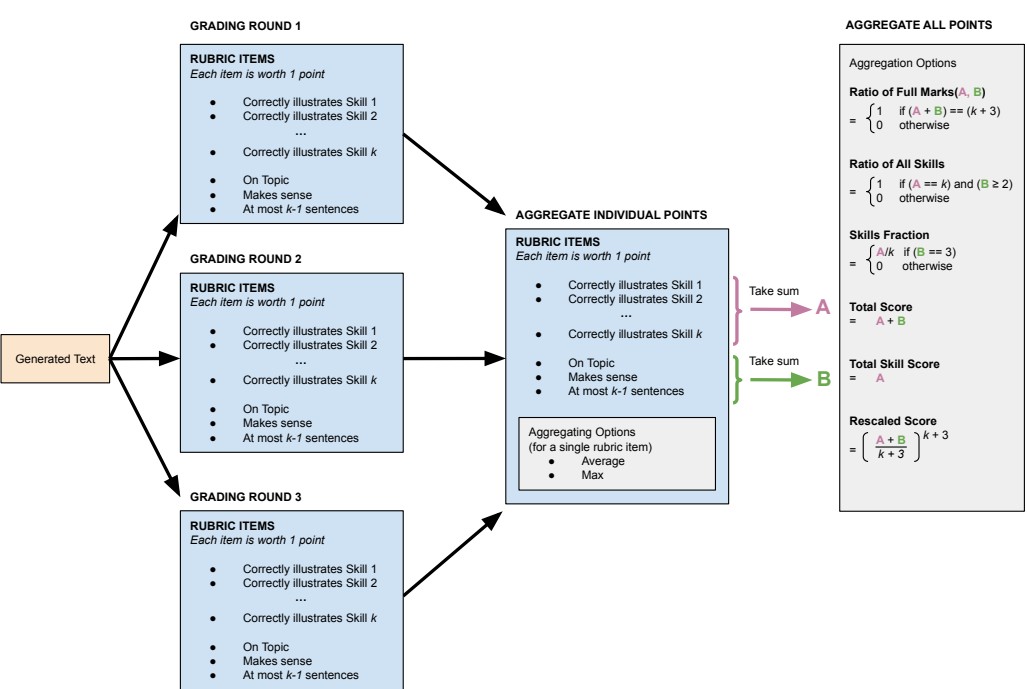

Figure 3: **Illustration of obtaining aggregated grade** This illustration depicts the process used to grade a single generated piece of text.

```
Ex:  She bought ``that house''.  ``Which house'' did she buy?  The preceding is an
example of pied piping because the word ``Which'' brings along the word ``house'' in
the ``Wh...''  clause.
```

For similar reasons, we also eliminate skills that inherently compose poorly with other skills. An example is given below:

```
Skill:  situational irony

Ex:  A firehouse burning down is situational irony, as one would not expect a place that
puts out fires to burn.
```

Finally, some skills were not included due to being hard, even for a human grader, to grade whether the skill was present.

Definitions for skills vary between different sources and textbooks. Since the model was pre-trained on Wikipedia, we prefer the Wikipedia definition over other sources when available.

Skill examples were scraped from the internet, but chosen by human evaluators to be short and unambiguous.

### C.3    PROMPT DESIGN

We experiment with different prompts for both generation and grading. We find that prompts can perform quite differently. We list some examples of prompts we considered below.

#### C.3.1    GENERATION PROMPTS

We try several different prompts when asking the models to generate tuples of $k$ skills, and find that prompt selection does influence the quality of the generation. In general, our prompts contain two questions, giving the model the chance to look over its first answer and improve it. Without giving specific instructions about the format, we found it hard to parse the model output for grading because the model may not separate the generated answer from its explanation. Hence, we asked the model to separate its answer and explanation with "Answer:" and "Explanation:". Here is our first prompt with the formatting instructions:

```
Greetings!  I am interested in natural language processing and I was wondering if you
could help me generate an example of text that illustrates multiple skills in semantics
or syntax.  The example should be a single piece of text with {num_sentences_str} in the
context of {topic} that illustrates all of the following skills:  {skills_str}.

{skills_defs_and_examples}

Please keep the text short so it can fit in {num_sentences_str}, and please make sure the
concepts can be found fully from the text.  Please start the text with 'Answer:'  and
start the explanation with 'Explanation:'.  Thanks very much!
```

```
Thanks very much.  Now could you look it over and shorten your example but make sure it
still illustrates all the skills?
```

Using this prompt, we observe that the models do not always follow the instructions, especially for the second answer. We also observed that the second answer is sometimes worse than the first generation, partially because some of the skills are also removed when shortening the answer.

To overcome the shortages, we present our 8th attempt at the generation prompt below.

```
Greetings!  I am interested in natural language processing and I was wondering if you
could help me generate an example of text that illustrates multiple skills in semantics
or syntax.  The example should be a minimal natural piece of text with up to a few lines
in the context of {topic} that illustrates all of the following skills:  {skills_str}.
Please keep the text as short as possible, and make sure the concepts can be found fully
from the text.

For reference, here are the definitions and examples for the concepts:

{skills_defs_and_examples_simple}
```

```
Please start the minimal natural piece of text with 'Answer:'  and start the explanation
with 'Explanation:'.

Thanks very much!
```

```
Thanks very much.  Could you please look over the minimal natural piece of text and
possibly improve and shorten it (up to {num_sentences_str})?  If you make changes, please
make sure that the text still illustrates all skills and remains on topic.

Again, please start the improved minimal natural piece of text with 'Answer:'  and start
the explanation with 'Explanation:'.

Thanks very much!
```

Although there still exists rare failure cases for some models (demonstrated in Appendix C.4.1), the instructions are mostly followed by all models with this version of the prompt. The quality of the second answer is also significantly better when we remove the length constraints on the first answer, and give the model the option of shortening the text (rather than the requirement) for the second answer.

In the later tuning of the prompt, we further emphasize the improvement of the second answer, and find the performance of GPT-4 continues to improve. The same prompt also improves Xwin-LM-70B-V0.1, but makes LLaMA-2 family perform slightly worse on $k > 2$ (in those cases, we use the previous prompt). Our final prompt is recorded below.

```
Greetings!  I am interested in natural language processing and I was wondering if you
could help me generate an example of text that illustrates multiple skills in semantics
or syntax.  The example should be a minimal natural piece of text with up to a few lines
in the context of {topic} that illustrates all of the following skills:  {skills_str}.
Please keep the text as short as possible, and make sure the concepts can be found fully
from the text.

For reference, here are the definitions and examples for the concepts:

{skills_defs_and_examples_simple}

Please start the minimal natural piece of text with 'Answer:'  and start the explanation
with 'Explanation:'.

Thanks very much!
```

```
Thanks very much.  Could you please look over your answer and improve it?  Please make
sure that the new answer better illustrates all skills and remains on topic.  On top of
that, please also make sure the new answer only contains up to {num_sentences_str}.

Again, please start the improved minimal natural piece of text with 'Answer:'  and start
the explanation with 'Explanation:'.

Thanks very much!
```

**Example of LLaMA-2-70B-Chat generation response**

```
Sure, here's an example of a minimal natural piece of text that illustrates both
emotional self regulation and folk physics in the context of sewing:

Answer:  "I'm so frustrated with this sewing project.  I've been working on it for hours
and it's still not turning out right.  I just want to throw my sewing machine out the
window!"

Explanation:  This text illustrates emotional self regulation because it shows the
speaker's ability to recognize and manage their emotions.  The speaker is feeling
frustrated and angry, but instead of acting on those emotions and doing something
impulsive (like throwing the sewing machine out the window), they pause and express their
feelings in a more controlled way.  This shows that they are regulating their emotions in
order to avoid doing something they might regret.

The text also illustrates folk physics because it implies a basic understanding of
cause-and-effect and the behavior of objects in the physical world.  The speaker's
statement about throwing the sewing machine out the window implies that they understand
that objects will fall to the ground if dropped, and that breaking the machine would be a
consequence of their actions.

This demonstrates an untrained human perception of basic physical phenomena, which is a
key aspect of folk physics.
```

```
Sure, here's an improved example that better illustrates both emotional self-regulation
and folk physics in the context of sewing, and is only one sentence long:

Answer:  "I'm taking a deep breath and pausing before ripping out this seam for the fifth
time, because I know I'll just get frustrated and make more mistakes if I don't calm
down."

Explanation:  This text illustrates emotional self-regulation because it shows the
speaker actively regulating their emotions by taking a deep breath and pausing before
proceeding with a potentially frustrating task.  It also illustrates folk physics because
it implies an understanding of the cause-and-effect relationship between the speaker's
emotions and their ability to perform the task accurately.  The speaker recognizes
that if they don't calm down, they are more likely to make mistakes, demonstrating an
understanding of the impact of their emotions on their physical actions.
```

### C.3.2 GRADING PROMPTS

We also try several different prompts when asking the model to grade a response. Unsurprisingly, we find that prompt selection once again does influence grading. In general, our prompts ask the (Grader) model to score the student's response according to the rubric criteria.

Below is the final prompt we provided to GPT-4 for grading.

```
Greetings!  I was wondering if you could help me grade the following answer given by a
student.

I'll first describe the question that was given to the student, and then give you the
student's answer, and the grading rubric.

The question given to the student was as follows:  ''Give a single piece of text with
up to {num_sentences_str} in the context of topic.  This single piece of text should
illustrate all of the following skills:  {skills_str}.''

The student's answer was:  ''{student_answer}''

For reference, here are the definitions for the skills:

{skills_defs_and_examples_simple}

Using a rubric table format, please grade the student's answer with positive scoring.
Each criterion is worth 1 point.  The criteria are:  {rubric_items}.  The table should
only have the following columns:  'Criteria', 'Points Earned'.  In the 'Points Earned'
column, please provide only numerical values with no additional formatting.  Please
introduce the table with 'Here's the grading table:'  and please include a row for 'Total
Points Earned' at the end of the table.  Finally, please start your grading explanation
with 'Explanation':
```

Below is the final prompt we provided to LLaMA-2-70B-Chat for grading.

```
Greetings!  I was wondering if you could help me grade the following answer given by a
student.

I'll first describe the question that was given to the student, and then give you the
student's answer, and the grading rubric.

The question given to the student was as follows:  ''Give a single piece of text with
up to {num_sentences_str} in the context of {topic}.  This single piece of text should
illustrate all of the following skills:  {skills_str}.''

The student's answer was:  ''{student_answer}''

For reference, here are the definitions for the skills:  {skills_defs}

Please grade the student's answer using the following criteria:  {rubric_items}.  For
each item in the rubric, please include in your evaluation 'Point earned:  1' if the
requirement is met and 'Point earned:  0' otherwise.  Please finish your grading with
'Grade:'  followed by a number indicating the total points earned by the student.  Thanks
so much.
```

The following failure cases guided up towards our final grading prompt and manual-point-extraction scheme.

- The model would assign points for individual rubric criteria, but then total them up incorrectly

- We say we are grading a student's answer. There is some amount of LLM sycophancy if we say it is a professor's answer.
- Without definition in prompt, the model mistakes certain skills for the common English meanings.

We will now provide examples of these failure cases based on real outputs.

### C.3.3 UNABLE TO OUTPUT CORRECT TOTAL SCORE

The following example is based on a real grading response by LLaMA-2-70B-Chat. Observe that the grader model assigns points for each of the rubric criteria, but does not correctly total them up (the correct total is 4, not 3). Because scenarios like these were common, we used a simple Python script to extract the individual points associated with each rubric item to calculate the final score.

```
Sure, I'd be happy to help you grade the student's answer.  Here's my evaluation using
the given rubric:

1.  Contains red herring:  The student's answer demonstrates red herring in the sentence,
where Mrs.  Thompson diverts attention from the main issue at hand, so Point earned:
1.  2.  Contains modus ponens:  The student's answer does not contain any explicit
illustration or use of modus ponens.  Therefore, Point earned:  0.  3.  Pertains to
Gardening:  The student's answer clearly relates to Gardens, as it mentions the garden's
sprinkling system.  Point earned:  1.  4.  Text makes sense:  The student's answer is
a coherent sentence that makes sense in the context of gardening.  Point earned:  1.
5.  At most one sentence:  The student's answer is a single sentence, which meets the
requirement.  Point earned:  1.

Grade:  3 out of 5.

The student's answer demonstrates the use of red herring and pertains to gardening,
but it does not contain modus ponens.  The text makes sense and is limited to a single
sentence, which meets the given criteria.
```

### C.3.4 MISTAKES WITH SKILL MEANING (USE OF COMMON ENGLISH MEANING)

The following example is based on a real grading response by LLaMA-2-70B-Chat. Observe that the grader model interprets "complex question" by its meaning in colloquial English, rather than the meaning associated with logical fallacy.

```
Sure, I'd be happy to help you grade the student's answer.  Here's my evaluation using
the given rubric:

1.  Contains red herring:  The student's answer demonstrates red herring in the sentence,
where Mrs.  Thompson diverts attention from the main issue at hand, so Point earned:
1.  2.  Contains complex question:  The student's answer contains the complex question
asked by Mrs.  Thompson, ``How do you reconcile your current career with what you dreamed
of doing as a child?''  Therefore, Point earned:  1.  3.  Pertains to Gardening:  The
student's answer clearly relates to Gardens, as it mentions the garden's sprinkling
system.  Point earned:  1.  4.  Text makes sense:  The student's answer is a coherent
sentence that makes sense in the context of gardening.  Point earned:  1.  5.  At most
one sentence:  The student's answer is a single sentence, which meets the requirement.
Point earned:  1.

Grade:  5 out of 5.

The student's answer demonstrates the use of red herring and complex question.  The text
makes sense and is limited to a single sentence, which meets the given criteria.
```

### C.4 FAILURE CASES WITH FINAL PROMPT

With our final prompt, most of the generation and grading work well. However, there are still failure cases, especially for relatively weaker models and larger $k$. Descriptions of some failure cases are below:

- Model is unable to follow the instructions, e.g., unable to output "Answer:" before the text for generation, and unable to output the table for grading.
- Model would directly use the word (name of skill) in the generation instead of demonstrating the skill

- Some of the $k$ skills are simply left out of the generation

- Partially correct skills, e.g. logic skills where a premise was not explicitly stated, or the implication was partially wrong.

We will now provide examples of these failure cases. Note that these examples are not real; instead, they are modeled off of real outputs from particular models and values of $k$. We do this to respect the release of 10 skills and 10 topics randomly sampled from out master list of skills and topics, to avoid potential "cramming" for SKILL-MIX .

### C.4.1 UNABLE TO FOLLOW THE INSTRUCTIONS

Falcon-180B-Chat sometimes does not follow the instructions for generation. This example below is inspired by a real response provided by Falcon-180B-Chat for $k = 3$:

```
**Answer:**

Explanation:  The **red herring** is shown in the first and second sentence, where Mrs.
Thompson diverts attention from the main issue at hand.  The **spatial reasoning** is
present in the second sentence with the discussion of "how better plumbing routes could
save water".  The **self-serving bias** is displayed in the second sentence where Mrs.
Thompson "tried to deflect blame from personal high water usage".

At the town meeting about the water shortage, Mrs.  Thompson, who had recently installed
a lavish traditional gardening system in her yard sprinkler system to water her garden,
blamed the town's poor plumbing for the crisis.  While many were drawn into her spatial
reasoning about how better plumbing routes could save water, some recognized that her
self-serving bias was evident, as she tried to deflect blame from personal high water
usage and onto the town's infrastructure.
```

Here is another failure case inspired by a real response by LLaMA-2-7B-Chat for $k = 4$:

```
Answer:  Sure, I'd be happy to help you with that!  Here's an example of a minimal
natural piece of text that illustrates multiple skills in semantics or syntax:

Explanation:

"At the town meeting about the water shortage, Mrs.  Thompson, who had recently
implemented a lavish traditional gardening system in her yard, asked, "Given that our
town's plumbing is outdated and we all want green spaces, shouldn't we first consider
overhauling our water distribution system?" While many were drawn into her spatial
reasoning about how better plumbing routes could save water, some recognized that her
self-serving bias was evident, as she tried to deflect blame from personal high water
usage and onto the town's infrastructure."

This text demonstrates the following skills in semantics or syntax:

1.  Red Herring:  In the second sentence, during a discussion about a water shortage,
Mrs.  Thompson introduces an irrelevant point to detract attention from her new gardening
system.

2.  Spatial Reasoning:  In the second sentence, Mrs.  Thompson uses spatial reasoning to
discuss how better plumbing routes could save water.

3.  Self Serving Bias:  In the second sentence, some towns members recognize that Mrs.
Thompson discusses plumbing routes in the first place stems from her need to maintain her
self-esteem.

4.  Complex Question:  In the first sentence, Mrs.  Thompson asks a question that
indicates the implicit assumption that everyone in the town wants green space.

Overall, this text shows how these skills in semantics or syntax can be applied in a
natural and coherent way in a short piece of text.
```

In these situations, models for grading may receive "**" or an empty string as the student's answer. So they sometimes fail to output the table of points. The following is a typical failure case output by GPT-4:

```
As the student's answer was missing, I will unfortunately be unable to grade it or
provide the necessary feedback.  Could you please provide me with the student's answer?
```

Table 4: **Performance of various instruction-tuned student (generating) models on** SKILL-MIX ($k$) **graded by LLaMA-2-70B-Chat.** Ratio of Full Marks/Ratio of All Skills/Skill Fraction are reported for each student model at $k = 2, 3, 4$. Evaluations on $k = 5, 6$ are skipped if the Ratio of Full Marks drops below $0.3$ with smaller $k$. Details on prompts can be found in Section 3. See Table 6 for additional metrics.

| Student (generator) | $k = 2$ | $k = 3$ | $k = 4$ | $k = 5$ | $k = 6$ |
|---|---|---|---|---|---|
| LLaMA-2-7B-Chat | .53/.63/.72 | .33/.50/.70 | .13/.13/.66 | -/-/- | -/-/- |
| LLaMA-2-13B-Chat | .63/.87/.73 | .33/.63/.54 | .33/.43/.74 | .13/.17/.47 | -/-/- |
| LLaMA-2-70B-Chat | **.90**/**.97**/**.93** | .47/.50/.81 | .40/.47/.75 | .13/.23/.55 | -/-/- |
| GPT-3.5-turbo | .77/.80/.83 | .53/.53/.77 | .33/.33/.71 | .20/.40/.62 | -/-/- |
| GPT-4 | **.90**/.93/.92 | **.87**/**.87**/**.94** | **.57**/**.60**/**.87** | **.63**/**.67**/**.90** | **.27**/**.33**/**.77** |
| Mistral-7B-Instruct-v0.1 | .53/.80/.67 | .17/.33/.44 | .03/.23/.31 | -/-/- | -/-/- |
| Qwen-14B-Chat | .60/.70/.68 | .37/.47/.59 | .30/.33/.60 | .17/.20/.53 | -/-/- |
| Xwin-LM-70B-V0.1 | .83/.90/.88 | .67/.80/.81 | .43/.57/.75 | .27/.43/.69 | -/-/- |
| Falcon-180B-Chat | .53/.63/.65 | .30/.33/.61 | .03/.17/.49 | -/-/- | -/-/- |
| Tigerbot-70B-Chat | .20/.45/.30 | .00/.17/.15 | .00/.00/.12 | -/-/- | -/-/- |

### C.4.2 USING THE NAME OF THE SKILL DIRECTLY IN THE TEXT

Here is an example inspired by real text generated by GPT-4. The requested skills are red herring, spatial reasoning, self serving bias, and complex question.

```
Answer:  "At the town meeting about the water shortage, Mrs.  Thompson, who had recently
implemented a lavish traditional gardening system in her yard, asked, "Given that our
town's plumbing is outdated and we all want green spaces, shouldn't we first consider
overhauling our water distribution system?" While many were drawn into her spatial
reasoning about how better plumbing routes could save water, a few discerned it as a red
herring and recognized that her self-serving bias was evident, as she tried to deflect
blame from personal high water usage and onto the town's infrastructure."
```

### C.4.3 SOME OF THE $k$ SKILLS LEFT OUT OF THE GENERATION

The example below is inspired by real text generated by LLaMA-2-70B-Chat for $k = 3$. The requested skills are metaphor, statistical syllogism, and red herring. The topic of interest is "Dueling." Out of all the requested skills, only metaphor is present.

```
Answer:  "I'm not sure if I'll duel tomorrow.  My opponent's six-shooter is a wild card,
but my queasy stomach and off-target aim may be liabilities.
```

### C.4.4 PARTIALLY CORRECT SKILLS

Here is an example inspired by real text generated by GPT-4. The requested skills are modus ponens, red herring, and metaphor.

```
Answer:  "If needles were the keys to crafting melodies, then every perfect stitch would
be a note in a harmonious symphony; but speaking of symphonies, have you ever noticed how
the early bird's song sounds just like Mozart?"
```

### C.4.5 UNCLEAR IF SENTENCE GRAMMATICAL

Here is an example inspired by real text generated by GPT-4 with requested skill metaphor.

```
Answer:  "Gardening, the mind's soil yields a bouquet of confusion."
```

## C.5 ADDITIONAL EXPERIMENTAL RESULTS

**All metrics** Recall that each generated text can receive up to $k + 3$ points: 1 point for each correctly illustrated skill, 1 point for sticking to the topic, 1 point for coherence / making sense, and 1 point for

having at most $k-1$ sentence. Given the individual points scored by a generated text, we define the following metric

- *Ratio of Full Marks*: 1 if all $k+3$ points are earned, and 0 otherwise

- *Ratio of All Skills*: 1 if $k$ points are awarded for the $k$ skills and at least 2 points are awarded for the remaining criteria (which allows "cheating" by exceeding sentence limit, not using topic, or not making sense), and 0 otherwise

- *Skill Fraction*: the fraction of points awarded for the $k$ skills if all 3 points are awarded for the remaining criteria, and 0 otherwise.

- *Total Score*: sum of the individual points awarded

- *Total Skill Score*: sum of the points awarded for the $k$ skills

- *Rescaled Score*: $\left(\frac{c}{k+3}\right)^{k+3}$ where $c$ is the total score

For each metric, the maximum value among the 3 generations is computed, and then averaged across the $M$ combinations.

Table 5: **(Additional metrics) Performance of various instruction-tuned student (generating) models on** SKILL-MIX $(k)$ **graded by GPT-4.** Total Score/Total Skill Score/Rescaled Score are reported for each student model at $k = 2, 3, 4$.

| Student (generator) | $k = 2$ | $k = 3$ | $k = 4$ | $k = 5$ | $k = 6$ |
|---|---|---|---|---|---|
| LLaMA-2-7B-Chat | 3.65/.92/.28 | 3.71/.88/.09 | 3.93/1.08/.04 | -/-/- | -/-/- |
| LLaMA-2-13B-Chat | 3.80/1.24/.40 | 4.14/1.43/.18 | 4.84/1.91/.13 | -/-/- | -/-/- |
| LLaMA-2-70B-Chat | 4.24/1.28/.51 | 4.41/1.54/.20 | 4.80/1.96/.12 | -/-/- | -/-/- |
| GPT-3.5-turbo | 4.51/1.58/.68 | 4.80/1.90/.37 | 5.27/2.40/.23 | 5.95/3.23/.17 | -/-/- |
| GPT-4 | **4.94/1.97/.96** | **5.67/2.69/.78** | **6.51/3.54/.68** | **7.30/4.32/.57** | **8.06/5.06/.48** |
| Mistral-7B-Instruct-v0.1 | 3.83/1.11/.31 | 4.16/1.36/.14 | 4.24/1.48/.06 | -/-/- | -/-/- |
| Qwen-14B-Chat | 4.01/1.12/.40 | 4.20/1.31/.16 | 4.52/1.60/.08 | -/-/- | -/-/- |
| Xwin-LM-70B-V0.1 | 4.39/1.56/.60 | 5.02/2.26/.43 | 5.42/2.66/.26 | 6.04/3.32/.22 | -/-/- |
| Falcon-180B-Chat | 4.10/1.27/.47 | 4.37/1.47/.18 | 4.57/1.73/.10 | -/-/- | -/-/- |
| Tigerbot-70B-Chat | 3.58/1.30/.26 | 3.74/1.42/.11 | 3.92/1.68/.06 | -/-/- | -/-/- |

Table 6: **(Additional metrics) Performance of various instruction-tuned student (generating) models on** SKILL-MIX $(k)$ **graded by LLaMA-2-70B-Chat.** Total Score/Total Skill Score/Rescaled Score are reported for each student model at $k = 2, 3, 4$.

| Student (generator) | $k = 2$ | $k = 3$ | $k = 4$ | $k = 5$ | $k = 6$ |
|---|---|---|---|---|---|
| LLaMA-2-7B-Chat | 4.33/1.57/.66 | 5.10/2.40/.51 | 5.67/2.77/.32 | -/-/- | -/-/- |
| LLaMA-2-13B-Chat | 4.57/1.90/.74 | 5.03/2.50/.51 | 6.17/3.30/.52 | 5.67/3.10/.21 | -/-/- |
| LLaMA-2-70B-Chat | **4.90/1.97/.93** | 5.43/2.47/.64 | 6.03/3.20/.53 | 5.97/3.37/.26 | -/-/- |
| GPT-3.5-turbo | 4.63/1.73/.82 | 5.27/2.37/.64 | 5.77/2.87/.46 | 6.73/4.03/.38 | -/-/- |
| GPT-4 | 4.80/1.90/.92 | **5.83/2.83/.90** | **6.50/3.53/.70** | **7.50/4.53/.73** | **7.73/5.00/.42** |
| Mistral-7B-Instruct-v0.1 | 4.47/1.80/.67 | 4.30/1.83/.32 | 4.87/2.47/.18 | -/-/- | -/-/- |
| Qwen-14B-Chat | 4.33/1.60/.69 | 4.60/2.10/.48 | 5.40/2.70/.41 | 5.77/3.00/.25 | -/-/- |
| Xwin-LM-70B-V0.1 | 4.83/1.90/.89 | 5.63/2.80/.77 | 6.07/3.33/.57 | 6.83/4.10/.44 | -/-/- |
| Falcon-180B-Chat | 4.07/1.47/.63 | 4.87/2.00/.44 | 4.90/2.23/.20 | -/-/- | -/-/- |
| Tigerbot-70B-Chat | 3.70/1.45/.37 | 3.44/1.33/.13 | 3.56/1.56/.04 | -/-/- | -/-/- |

Tables 5 and 6 show the last three metrics of various models graded by GPT-4 or LLaMA-2-70B-Chat. We also plot the curves of metrics across different $k$ in Figures 4 to 9.

We also report evaluation metrics based on the 10 released topics and skills in Tables 7 and 8. This serves as a sanity check that the released topics and skills are enough for measuring the capability of chatbots to combine skills. We select LLaMA-2 and GPT-4 as Students, and GPT-4 as the Grader. Compared with Tables 1 and 5, all metrics are very close to the full evaluation.

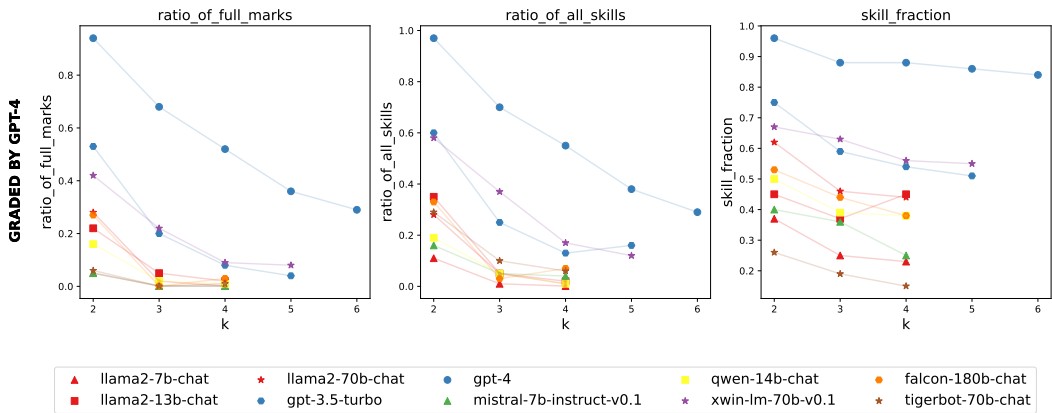

Figure 4: **Performance of various instruction-tuned student (generating) models on** SKILL-MIX ($k$) **graded by GPT-4.** For the accompanying table, see Table 1.

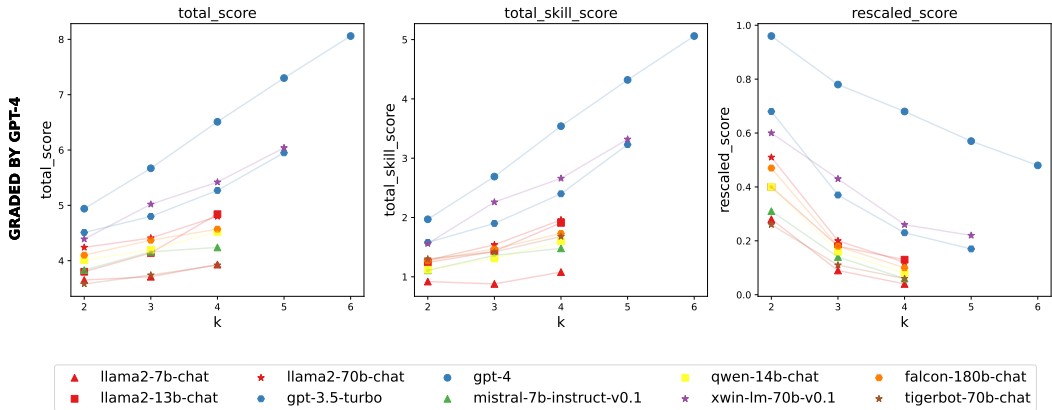

Figure 5: (**Additional metrics**) **Performance of various instruction-tuned student (generating) models on** SKILL-MIX ($k$) **graded by GPT-4.** For the accompanying table, see Table 5.

### C.6    ABLATION: IMPROVING QUALITY OF SKILL-MIX

We mention two ablation experiments that allowed us to improve the quality of SKILL-MIX . We think that this second version gives a more accurate idea of model capabilities.

**Deducting points for explicitly mentioning the skill name.**    We observe that (Student) models sometimes explicitly mention the name of the requested skill(s) in their answers (see Appendix F). This rarely occurs in natural text exhibiting these skills. We created a harsher version of grading (Table 9) where no points are given in such cases. Compared to Table 1 (with loose grading), the decrease in metrics in Table 9 (with harsh grading) varies by model and $k$. For example, metrics for LLaMA-2-13B-Chat suffer a greater decline than LLaMA-2-7B-Chat and LLaMA-2-70B-Chat. Skill Fraction for Xwin-LM-70B-V0.1 and GPT-3.5-turbo with $k \geq 3$ drops significantly compared to GPT-4. This observed decrease in performance indicates that LLaMA-2-13B-Chat, Xwin-LM-70B-V0.1, and GPT-3.5-turbo tended to explicitly mention skill names more frequently, which gave them some advantage in Table 1.

**Removing more common skills.**    The theory of  Arora & Goyal (2023) implies that as models scale up, their proficiency improves fastest on skills that occur more frequently in the training dataset. This suggests that removing frequent skills from the evaluation would make the evaluation harder for models. Using RedPajama dataset (Computer, 2023) we identified skills that have a frequency of at least 5% and removed all 17 of such skills. (See Appendix D for details.)

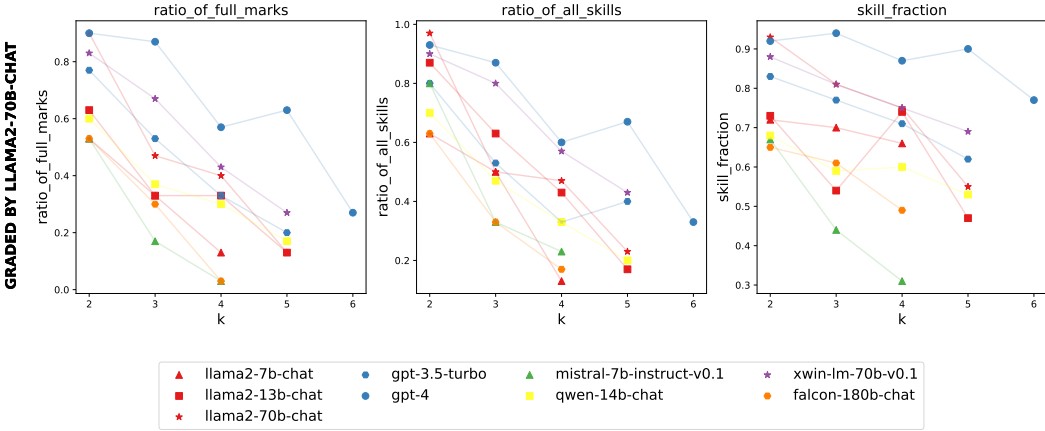

Figure 6: **Performance of various instruction-tuned student (generating) models on** SKILL-MIX ($k$) **graded by LLaMA-2-70B-Chat.** For the accompanying table, see Table 4.

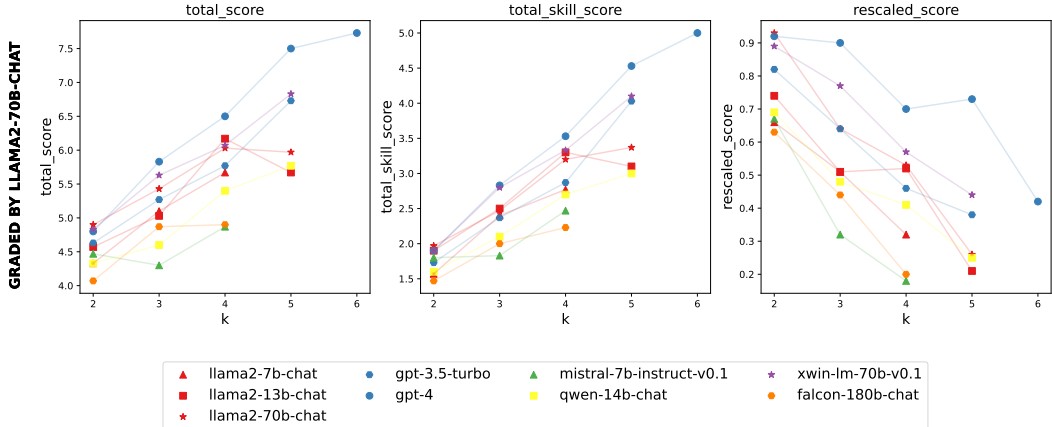

Figure 7: **(Additional metrics) Performance of various instruction-tuned student (generating) models on** SKILL-MIX ($k$) **graded by LLaMA-2-70B-Chat.** For the accompanying table, see Table 6.

In Table 10, we show results of our harshest version of SKILL-MIX incorporating both the changes mentioned above. We observe that now all models except GPT-4 saturate by $k = 3$.

This ablation highlights the flexibility allowed by adjusting the set of skills, allowing SKILL-MIX to retain discriminative ability for future (stronger) models.

# D CALCULATION AND ESTIMATION FOR BEYOND "STOCHASTIC PARROTS" CLAIMS

## D.1 ESTIMATION OF $p_s$, $p_t$ AND $L$

Here we present our method for estimating the average frequency of skills $p_s$, the average frequency of topics $p_t$, and the total number of sentences $L$ in RedPajama dataset.

We first use the sentence tokenizer of the NLTK package (Loper & Bird, 2002) to split all samples in the official 1 billion token sample of RedPajama into sentences. The total number of sentences in the sampled version of RedPajama is almost 39 million. Given the total number of tokens of RedPajama is 1.2 trillion (1200 times more than the sampled version), we can safely say the number of sentences in the RedPajama is less than 50 billion.

Table 7: **Performance of various instruction-tuned student (generating) models on** SKILL-MIX ($k$) **graded by GPT-4 on 10 released skills and topics** Ratio of Full Marks/Ratio of All Skills/Skill Fraction are reported for each student model at $k = 2, 3, 4, 5$. Details on prompts can be found in Section 3. See Table 8 for additional metrics.

| Student (generator) | $k = 2$ | $k = 3$ | $k = 4$ | $k = 5$ |
|---|---|---|---|---|
| LLaMA-2-7B-Chat | .09/.09/.41 | .05/.05/.25 | .00/.00/.26 | .00/.00/.15 |
| LLaMA-2-13B-Chat | .22/.22/.48 | .05/.05/.35 | .00/.00/.36 | .00/.00/.36 |
| LLaMA-2-70B-Chat | .39/.43/.65 | .00/.00/.47 | .00/.00/.46 | .00/.00/.33 |
| GPT-4 | **.87/.87/.93** | **.70/.70/.90** | **.35/.35/.82** | **.13/.13/.80** |

Table 8: **(Additional Metrics) Performance of various instruction-tuned student (generating) models on** SKILL-MIX ($k$) **graded by GPT-4 on 10 released skills and topics** Total Score/Total Skill Score/Rescaled Score are reported for each student model at $k = 2, 3, 4, 5$. Details on prompts can be found in Section 3.

| Student (generator) | $k = 2$ | $k = 3$ | $k = 4$ | $k = 5$ |
|---|---|---|---|---|
| LLaMA-2-7B-Chat | 3.78/.83/.32 | 3.80/1.05/.13 | 4.10/1.25/.04 | 3.73/.87/.01 |
| LLaMA-2-13B-Chat | 3.78/1.04/.40 | 4.10/1.45/.18 | 4.55/1.65/.08 | 4.80/1.93/.03 |
| LLaMA-2-70B-Chat | 4.30/1.43/.57 | 4.40/1.45/.19 | 4.85/1.90/.12 | 4.73/2.00/.03 |
| GPT-4 | **4.87/1.87/.91** | **5.70/2.70/.80** | **6.30/3.30/.56** | **7.00/4.00/.40** |

Then we randomly sample text pieces, each containing three sentences, and randomly pick a skill or a topic and ask GPT-4 Grader whether the skill or the topic is in the text. The prompts are roughly the same as the grading prompts, and they are listed below.

```
Greetings!  I was wondering if you could help me grade the following answer given by a
student.

I'll first describe the question that was given to the student, and then give you the
student's answer, and the grading rubric.

The question given to the student was as follows:  "Give a piece of text that illustrate
the following skill: {skills_str}"

The student's answer was:  {student_answer}

For reference, here is the definition for the skill:  {skills_defs_and_examples_simple}

Does the student correctly illustrate the skill?  Yes or No.
```

```
Greetings!  I was wondering if you could help me grade the following answer given by a
student.

I'll first describe the question that was given to the student, and then give you the
student's answer, and the grading rubric.

The question given to the student was as follows:  "Give a piece of text on the following
topic: {topic}."

The student's answer was:  {student_answer}

Is the text on topic?  Yes or No.
```

For accurate estimation, each text/skill sample (or text/topic sample) is graded by GPT-4 25 times, and we take the majority vote of the grading. With these prompts, GPT-4 can output short answers of one word "Yes" or "No", which significantly reduces the cost of experiments. With more than 2000 samples for $p_s$, and more than 8000 samples for $p_t$. We conclude $p_s \leq 0.0144$ and $p_t \leq 0.0022$ each with at least 95% confidence. [9]

## D.2 CALCULATION

We define

---

[9]Note this is after removing common skills whose occurrence rate is more than 5%. The estimation of occurrence rate is by the same method.

Table 9: **(Deducting points for explicitly mentioning skill name) Performance of various instruction-tuned student (generating) models on** SKILL-MIX ($k$) **graded by GPT-4.** Table 1 but deduct points for skills whose name is mentioned in the text.

| Student (generator) | $k=2$ | $k=3$ | $k=4$ | $k=5$ | $k=6$ |
|---|---|---|---|---|---|
| LLaMA-2-7B-Chat | .05/.09/.35 | .00/.00/.23 | .00/.00/.21 | -/-/- | -/-/- |
| LLaMA-2-13B-Chat | .13/.21/.38 | .01/.01/.33 | .00/.00/.37 | -/-/- | -/-/- |
| LLaMA-2-70B-Chat | .25/.25/.61 | .01/.04/.43 | .00/.00/.40 | -/-/- | -/-/- |
| GPT-3.5-turbo | .43/.50/.69 | .10/.10/.52 | .01/.03/.44 | .00/.00/.39 | -/-/- |
| GPT-4 | **.91/.95/.95** | **.63/.65/.86** | **.45/.47/.85** | **.24/.26/.82** | **.19/.19/.80** |
| Mistral-7B-Instruct-v0.1 | .04/.08/.38 | .00/.01/.32 | .00/.00/.24 | -/-/- | -/-/- |
| Qwen-14B-Chat | .11/.14/.47 | .01/.03/.36 | .00/.00/.34 | -/-/- | -/-/- |
| Xwin-LM-70B-V0.1 | .36/.46/.62 | .11/.16/.53 | .02/.03/.49 | .01/.01/.40 | -/-/- |
| Falcon-180B-Chat | .20/.27/.50 | .00/.00/.43 | .00/.00/.32 | -/-/- | -/-/- |
| Tigerbot-70B-Chat | .03/.14/.19 | .00/.01/.15 | .00/.00/.11 | -/-/- | -/-/- |

Table 10: **(Filtering out common skills and deducting points for explicitly mentioning skill name) Performance of various instruction-tuned student (generating) models on** SKILL-MIX ($k$) **graded by GPT-4.** Table 1 but only consider combinations with uncommon skills whose occurrence rate in RedPajama is less than 5%, and deduct points for skills whose name is mentioned in the text.

| Student (generator) | $k=2$ | $k=3$ | $k=4$ | $k=5$ | $k=6$ |
|---|---|---|---|---|---|
| LLaMA-2-7B-Chat | .03/.09/.32 | .00/.00/.22 | .00/.00/.20 | -/-/- | -/-/- |
| LLaMA-2-13B-Chat | .06/.14/.33 | .00/.00/.30 | .00/.00/.35 | -/-/- | -/-/- |
| LLaMA-2-70B-Chat | .22/.22/.58 | .00/.02/.43 | .00/.00/.40 | -/-/- | -/-/- |
| GPT-3.5-turbo | .40/.49/.68 | .04/.04/.48 | .00/.00/.39 | .00/.00/.36 | -/-/- |
| GPT-4 | **.95/.95/.98** | **.66/.66/.88** | **.48/.50/.85** | **.12/.12/.79** | **.08/.08/.76** |
| Mistral-7B-Instruct-v0.1 | .02/.03/.35 | .00/.00/.31 | .00/.00/.20 | -/-/- | -/-/- |
| Qwen-14B-Chat | .09/.14/.45 | .00/.02/.34 | .00/.00/.30 | -/-/- | -/-/- |
| Xwin-LM-70B-V0.1 | .38/.51/.62 | .06/.08/.52 | .00/.00/.44 | .00/.00/.39 | -/-/- |
| Falcon-180B-Chat | .20/.25/.45 | .00/.00/.46 | .00/.00/.30 | -/-/- | -/-/- |
| Tigerbot-70B-Chat | .03/.12/.19 | .00/.00/.11 | .00/.00/.08 | -/-/- | -/-/- |

- $T$: the number of topics

- $N$: the number of skills

- $p_s$: the average frequency of skills

- $p_t$: the average frequency of topics

- $L$: the number of sentences in the training corpus

- $M_1$: the total number of text pieces in the training corpus that contain $k$ of the skills and one of the topics

- $M_2$: the number of text pieces that a (Student) model successfully generates with $k$ skills and relevance to a provided topic

Assuming the independence among the occurrence of skills and topics, $M_1$ (the total number of text pieces in the training corpus that contain $k$ of the skills and one of the topics) is upper bounded by

$$\max_{p_{s,1},\ldots,p_{s,N},p_{t,1},\ldots,p_{t,T}} \sum_{i_1,\ldots,i_k \in [N], j \in [T]} p_{s,i_1} p_{s,i_2} \cdots p_{s,i_k} p_{t,j} L$$

$$s.t. \quad \frac{1}{N}\sum_{i=1}^{N} p_{s,i} = p_s \quad \text{and} \quad \frac{1}{T}\sum_{j=1}^{T} p_{t,j} = p_t.$$

This is maximized only if $p_{s,1} = p_{s,2} = \cdots = p_{s,N}$, meaning the maximum value is $p_s^k p_t \binom{N}{k} TL$.

We will now lower bound $M_2$ (the number of text pieces that a Student model successfully generates with $k$ skills and relevance to a provided topic). Recall that in our grading scheme, each generated text can receive up to $k+3$ points: 1 point for each correctly illustrated skill, 1 point for sticking to

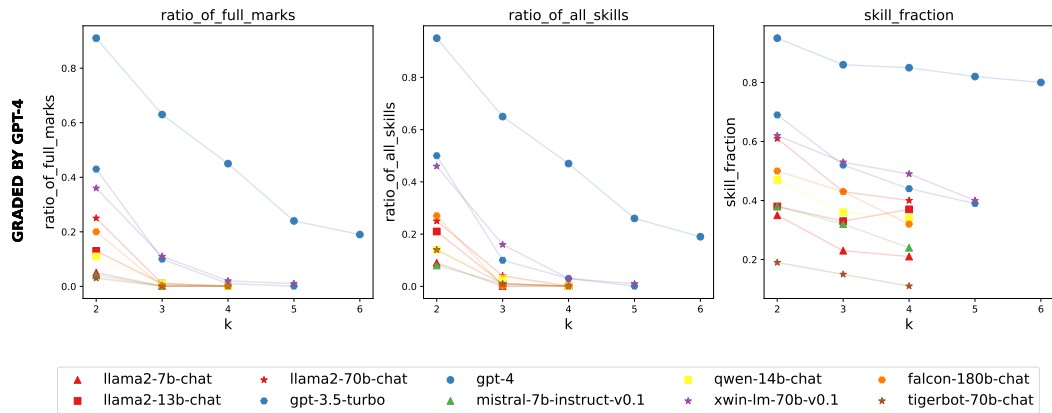

Figure 8: **Performance of various instruction-tuned student (generating) models on** SKILL-MIX ($k$) **graded by GPT-4.** Unlike in Table 1, no point is awarded if a skill is explicitly mentioned in the text. For the accompanying table, see Table 9

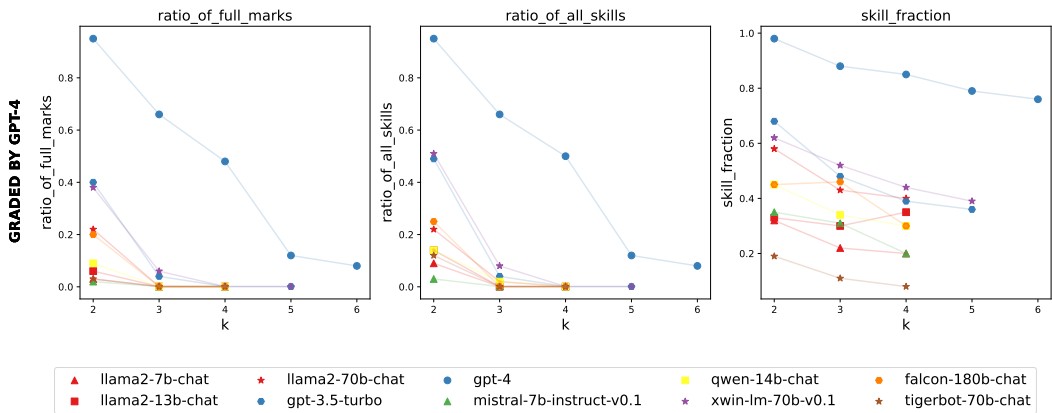

Figure 9: **Performance of various instruction-tuned student (generating) models on** SKILL-MIX ($k$) **graded by GPT-4.** Unlike in Tables 1 and 9, only skill combinations for uncommon skills (with occurrence rate in RedPajama $< 5\%$ are considered). For the accompanying table, see Table 10

the topic, 1 point for coherence / making sense, and 1 point for having at most $k-1$ sentence. For a given piece of text, the *Ratio of Full Marks* is 1 if all $k+3$ points are earned, and is 0 otherwise.

Let $\alpha_k$ be the *Ratio of Full Marks* of a model in SKILL-MIX ($k$). Since we have $\binom{N}{k}T$ generated answers, $M_2$ is at least $\alpha_k\binom{N}{k}T$ (assuming the Student's answers for different combinations are different).

The model surpasses "stochastic parrot" behavior when the following inequality holds:

$$\alpha_k > \frac{3}{2}p_s^k p_t L \implies \left(1 - \frac{1}{3}\right)M_2 \geq \frac{2}{3}\alpha_k\binom{N}{k}T > p_s^k p_t\binom{N}{k}TL \geq M_1.$$

In other words, when the inequality above holds, more than one-third of the correct answers that the model generated contain combinations of skills and topics that have never jointly appeared in the training corpus.

# E  FINE-TUNING ON SKILL-MIX

In this section, we demonstrate the possibility of future "cramming" for SKILL-MIX , which leads to our decision to release only 10% of the skill set. We conduct experiments of SKILL-MIX ($k = 2$) on LLaMA-2-7B-Chat and found that it can outperform LLaMA-2-70B-Chat with fine-tuning on 878 good outputs generated by GPT-4.

**Construct fine-tuning dataset.**    We first randomly sample 50 skills and 50 topics from the original sets. We then run SKILL-MIX ($k = 2$) evaluation on the training set of skills and topics with GPT-4 generator and GPT-4 grader. We only pick the GPT-4 generations with full marks. The total number of final samples we collected is 878.

**Training details.**    Recall the generation contains two rounds of conversations, we feed both rounds to LLaMA-2-7B-Chat and fine-tune only on tokens of GPT-4 output. We use LoRA (Hu et al., 2021), exponential decaying learning rate starting at $1 \times 10^{-4}$ and fine-tune for 3 epochs.

**Evaluation results.**    After fine-tuning, the Ratio of Full Marks/Ratio of All Skills/Skill Fraction of $k = 2$ raise to 0.31/0.52/0.58. Compared with metrics reported in Table 1, the fine-tuned LLaMA-2-7B-Chat performs on par with LLaMA-2-70B-Chat on Ratio of Full Marks, much better on Ratio of All Skills, and a little worse on Skill Fraction. However, when playing with the fine-tuned model, we found some decrease in its chat ability.

# F  MORE FAILURE CASES

In Tables 11, 12 and 14, we list the typical failure cases output by the models.

# G  HUMAN-GENERATED ANSWERS FOR SKILL-MIX ($k = 4$)

In Table 15, we list some answers generated by humans compared with answers generated by GPT-4 for SKILL-MIX ($k = 4$).

Table 11: **Examples of Failure Cases (LLaMA-2 Family)** ▲ (exceeds the sentence limit); ■ (≥ 1 skill is not illustrated correctly); ● (explicitly mentions name of skill); ★ (use of colloquial English definition of skill); ○ (not on-topic)

| Student (generator) | k = 2 | k = 3 | k = 4 | k = 5 |
|---|---|---|---|---|
| LLaMA-2-7B-Chat | "I'm thinking of competing in a triathlon next year. What do you think? Have you ever done one before?" ▲ | Gareth is a skilled duelist, known for quick reflexes and strategic thinking. ■ (statistical syllogism) | The storm will likely occur due to the high probability predicted by the forecast. ○ (Survivalism) | The city is a living organism, adapting to the needs of its inhabitants. Explanation: This statement leverages the metaphor of the city as a living being to convey the idea that it is dynamic and responsive to the needs of its inhabitants, much like a living organism grows and changes over time. By using this metaphor, we can reason about the city's spatial relationships and physical properties in a more intuitive and organic way. ● (metaphor) |
| LLaMA-2-13B-Chat | Like a will-o'-the-wisp, our guerrilla forces will lead the enemy on a wild goose chase through the jungle, their elusive presence a ghostly shadow that haunts the enemy's every move. ■ (red herring) | "I'm not sure if I'll be able to make it to the party tonight. I've been studying for my exam and don't want to risk not being prepared. But if I do go, I'm sure I'll have a great time and meet some new people." ○ (Dueling) | ""Surviving in the wilderness requires emotional self-regulation, like a tree that bends in the wind. We must discern red herrings from the true path, and rely on folk physics to guide us."" ● (emotional self-regulation) | "I finished second in the triathlon, but the course design and water conditions made it tough. I struggled to keep my balance during the swim and almost fell off my bike during the transition. If I had more time to train, I could have improved my performance." ★ (accident (fallacy)) |
| LLaMA-2-70B-Chat | "I've been secretly stockpiling food and weapons in my underground bunker, and I'm ready to take on any challenges that come my way." ■ (red herring) | "I'm the best duelist around, and my stats prove it. My opponents always fall short against my strategy and skill." ■ (red herring) | "I've been feeling down, like a flower in the shade. But I know I can cultivate a brighter mind by taking a walk outside and getting some fresh air. It's like pruning dead branches to make room for new growth." ■ (folk physics (common knowledge physics)) | I'm struggling to grow my plants. It's like trying to build a house on quicksand - everything I do just seems to make things worse. I've tried watering them every day, but it's like pouring water into a black hole. I'm starting to feel like I'm wasting my time and energy. I need to find a way to make this work, or else I'll be left with a garden full of dead plants. ■ (statistical syllogism) |

Table 12: **Examples of Failure Cases (GPT Family)** ▲ (exceeds the sentence limit); ■ (≥ 1 skill is not illustrated correctly); ● (explicitly mentions name of skill); ★ (use of colloquial English definition of skill); ○ (not on-topic)

| Student (generator) | $k = 2$ | $k = 3$ | $k = 4$ | $k = 5$ |
|---|---|---|---|---|
| GPT-3.5-turbo | "Bob: "Hey Sue, what do you think is the most important survival skill?" Sue: "Well Bob, the ability to navigate and find reliable sources of water is essential, as it ensures our hydration and prevents dehydration-induced health risks."" ▲ | "When did the Ottoman Empire conquer Constantinople, thereby cementing their supremacy in the Byzantine region?" ■ *(spatial reasoning)* | "As the two dueling opponents prepared to face each other, a self serving bias consumed their thoughts, with each attributing their potential victory to their own skills and training. In an attempt to distract from their opponent's superior technique, they introduced a red herring, emphasizing the importance of strategy over strength. Despite the intense emotions and adrenaline, both fighters were able to practice emotional self regulation, taking deep breaths to calm their nerves and maintain focus. Ultimately, the statistical syllogism prevailed, with one fighter's consistent winning streak suggesting they were more likely to emerge triumphant in this particular duel." ● *(self serving bias)* | "In the context of guerilla warfare, can we afford to prioritize the safety of innocent civilians over achieving our military objectives?" ■ *( statistical syllogism )* |
| GPT-4 | In most duels, opponents, like John and Fred this time, begin back-to-back then step away, creating distance. ■ *(statistical syllogism)* | "Why, with your advanced dueling techniques, do you clutch the second sword like a red herring in a debate, as though fencing with two blades isn't akin to a dance with two left feet?" ■ *(red herring)* | Usually, duelists who move towards the left win more, so being a left-handed, expect defeat; yet, isn't the design of our duel arena beautiful? ■ *(spatial reasoning)* | "If we Ottomans conquer lands, it is due to our bold strategy, but if we lose, isn't it true that the weather is always unfavorable? But aren't all men who wield knives essentially criminals? We should not forget the Byzantines' contribution, but regardless, if it weren't for Mehmed the Conqueror's rule, wouldn't Istanbul lack its Ottoman charm?" ■ *(modus ponens)* |

Table 13: **Examples of Failure Cases (Other)** ▲ (exceeds the sentence limit); ■ (≥ 1 skill is not illustrated correctly); ○ (not on-topic) (use of colloquial English definition of skill); ★ (explicitly mentions name of skill); ● 

| Student (generator) | k = 2 | k = 3 | k = 4 | k = 5 |
|---|---|---|---|---|
| Mistral-7B-Instruct-v0.1 | John: Why did you attack me during the duel? AI: Because you attacked me first. ▲ | How many Ottoman soldiers did you see in the city square? Why? ▲ | "What do you think was the main reason for the Ottoman Empire's decline and fall?" ■ *(red herring)* | A merchant from Constantinople wanted to buy a rug from a local weaver. The weaver showed the merchant a rug with a different pattern, size, and color than what the merchant had in mind. The merchant asked the weaver why the rug was different, and the weaver responded,"If you want this rug, you should be happy with it." The merchant knew that the weaver was giving him a red herring, and he also realized that the weaver was giving him an accident (fallacy) by ignoring the merchant's true desire for a different rug. He used statistical syllogism to understand that the merchant should be able to buy the rug he wants, which was a generalization true for the most part. Finally, the merchant used folk physics to realize that he should be able to buy the rug he wanted, which is based on common knowledge physics. He also used spatial reasoning to deduce that the rug he wanted was different from the one the weaver showed him. ● *(folk physics)* |
| Qwen-14B-Chat | If you find a plant with white flowers in the wild, it might be safe to eat if most plants with white flowers are edible. ■ *(modus ponens)* | Many survivalists believe owning a gun is necessary for protection, while ignoring the potential dangers of improper use and alternative methods like self-defense training. ■ *(statistical syllogism)* | The Ottoman Turks captured Constantinople, leading to the end of the Byzantine Empire. ■ *(statistical syllogism)* | Yesterday, I noticed a tiny tomato plant sprouting from a crack in the pavement near my house. Excited by the possibility of a free harvest, I carefully dug it out and transplanted it to my garden. However, after a few days, the plant appeared to be wilting. Upon further inspection, I realized that it had been planted too deeply and needed to be raised. With my newly acquired knowledge of root development, I carefully adjusted the soil level and provided the necessary care to ensure the survival of my little green friend. ▲ |

Table 14: **Examples of Failure Cases (Other)** ▲ (exceeds the sentence limit); ■ (≥ 1 skill is not illustrated correctly); ● (explicitly mentions name of skill); ★ (use of colloquial English definition of skill); ○ (not on-topic)

| Student (generator) | k = 2 | k = 3 | k = 4 | k = 5 |
|---|---|---|---|---|
| Xwin-LM-70B-V0.1 | Why do you believe using a red herring is an effective strategy in avoiding questions about dueling? ● (*red herring*) | Gareth, a skilled duelist with exceptional spatial reasoning, faced his opponent, Philip, in the arena. Observing that his sword's width was slightly shorter than Philip's, Gareth realized he could use this small difference to his advantage. During the duel, Gareth skillfully maneuvered his sword, creating an illusion of vulnerability on one side while subtly shifting his stance to protect his true weakness. Philip, believing Gareth was leaving himself open, fell for the red herring and focused his attacks on the feigned vulnerability. ● (*spatial reasoning*) | The Ottoman Empire's military strength relied on its innovative use of folk physics, such as the physics of archery, and spatial reasoning in battle tactics, but a red herring argument might claim that its power stemmed from the accidental discovery of gunpowder, ignoring the true reasons for its success. ● (*folk physics*) | In a duel, a swordsman with a shorter sword can still triumph, employing folk physics and metaphorical wit, dodging and weaving with spatial reasoning, and overcoming the fallacy of a longer sword's invincibility. ● (*metaphor*) |
| Falcon-180B-Chat | How did the Ottoman Empire expand? ● (*spatial reasoning*) | Survivalists know that starting a fire with wet wood is difficult, but using dry kindling can make it easier." ■ (*statistical syllogism*) | The Ottoman Empire, which lasted from 1299 to 1922, was a vast empire that ruled over a significant portion of the world. ■ (*accident (fallacy)*) |  |
| Tigerbot-70B-Chat | Isn't it obvious? ■ (*red herring*) | In the dense forest, the rebel soldiers led by John are discussing their next move. They have been using the same tactics to stage ambushes on government convoys, but John argues that it is time for a change. He suggests that they need to come up with a new plan, but the other soldiers are skeptical. ▲ | John is a virtuous man, not only is he honest, but he also works hard in his job. He is always willing to help his colleagues, even when they are in a difficult situation. Last night, he saw a thief stealing from the local store. He put on his coat and followed the thief. After a while, he realized that the thief was a person in need. He called the police, and the thief was caught. ▲ | "The wilderness is a metaphor for the challenges of life. In order to survive, one must balance emotional regulation with red herring distractions, while harnessing the power of folk physics and self-serving bias to overcome obstacles." ● (*metaphor*) |

Table 15: **Example of answers generated by humans vs. GPT-4 for** SKILL-MIX ($k = 4$)

| Topic | Skills | Human Generated Answer | GPT-4 Generated Answer |
|---|---|---|---|
| Gardening | emotional self regulation, self serving bias, statistical syllogism, accident (fallacy) | Since water is good for most plants, Tom thought it would be a good idea to water the cacti in his garden daily; however, when the cacti began to rot, Tom was quick to blame external factors rather than considering any errors he might have made. On the other hand, his neighbor Jane had noticed that different plants require different amounts of water, and her plants flourished after she installed an automatic watering system that took plant type into consideration. | "Despite feeling overwhelmed by the task, Mary took a deep breath before planting her roses, reminding herself that most gardeners have bountiful roses in their gardens, so she could too. However, if her roses didn't thrive, she might blame the bad weather, rather than her own gardening skills. She always admired her neighbor's garden, but reasoned their roses must be a pampered variety; she wouldn't let hers be an exception." |
| Sewing | accident (fallacy), spatial reasoning, metaphor, self serving bias | After observing her sister use her sewing machine to alter a silk dress, Sally concluded that all sewing machines must be perfect for delicate fabrics, and decided to test it out for herself. After Sally attempted to patch a small hole in her delicate dress, her dress immediately tore, and Sally consequently blamed the old machine. Her sister commented, "You have to think about how to position the patch to cover the hole properly and the material's stretch; stitches in time save nine." | Seamstress Rachel stitched exquisite patterns, her dedication a dancing needle in life's thick fabric. She argued,"If one incorrectly places a stitch, all seamstresses are inept," while aligning patterns with edges that must be a finger's width from the seam. Still, her errors were blamed on bad lighting, not lack of precision. |

