# OpenReview forum: "SKILL-MIX: a Flexible and Expandable Family of Evaluations for AI Models"
_ICLR.cc/2024/Conference — ICLR 2024 poster_

### Official Review · Reviewer_VgUg · 2023-10-24

**Soundness:** 3 good
**Presentation:** 2 fair
**Contribution:** 3 good
**Rating:** 5
**Confidence:** 3

**Summary:**

The authors present Skill-Mix, an evaluation for LLMs which utilizes skill combinations to evaluate LLM capability on likely unseen tasks. They demonstrate that some of their results contradict existing results on popular LLM evaluations.

**Strengths:**

**********************Motivation:********************** One of the main motivations of the paper is good: LLM evaluations can be trained for so it makes sense to do some combinatorial benchmark which requires combining skills in a way that’s likely unseen in any training corpus.

**************Method:************** The idea of skill combination as an LLM evaluation is good, and to the best of my knowledge (i’m not very familiar with LLM benchmarks), novel. It is definitely harder to expect knowledge of the benchmark to be contained in the training corpora of LLMs a priori.

******************Dataset:****************** It’s good idea to release only 10% of the skills and topics for evaluation to prevent overfitting/training on the testing criteria.

****************Results:**************** I’m glad the paper demonstrates that this evaluation is not so correlated with other existing benchmarks, as to prove that it is testing something different from the others with respect to LLM capability.

**Weaknesses:**

****************************Skill picking:**************************** Section 3.1 (and the appendix) describe how the authors pick the skills by hand based on textbooks on logical reasoning, rhetoric, and theory of mind. But what is the justification for using these types of skills? For example, i’m sure there are many skills not related to any of those topics which still require some sort of intelligent capability to combine meaningfully. The authors should explain this better.

**********************Evaluation:**********************

- One minor issue is that I don’t think the models picked necessarily need to be instruction-finetuned as regular, non-instruction finetuned language models are still able to follow instructions in the prompt. In fact, given that some recent work has demonstrated that instruction-finetuning on smaller models reduces their ability to produce as diverse of a response distribution, it could artificially hurt the performance of the selected open-source models in evaluation due to the selected topics occurring with a Google n-gram probability of ~1e-6 and them likely being fine-tuned on a much smaller dataset that may not include the selected topics/skills.
- What is the variance among the automatically graded scores? It is stated that human variance is higher than the automatic ones, but neither seem to be presented anywhere.
- The analysis of Skill-Mix vs other benchmarks could be more comprehensive — it would be interesting to see ******************correlation scores****************** against other benchmarks to see how skill-mix differs overall wrt individual LLM rankings.

********Writing + Clarity + Motivation:********

- Intro first paragraph: A missing “yet” between the 2nd and third sentence, otherwise these two sentences next to each other are a bit confusing.
- Intro: “(including for some public models such as LLaMA-2)” this could be put next to “size of training corpora” as right now it’s directly after mentioning closed-source LLMs which makes the parenthesis seem confusing.
- Intro: Why is a specific opinion from Hinton used as motivation for desiderata for the authors’ benchmark? While Hinton is an important figure in the field, a good justification should come from (1) an intuitive explanation of the problem, (2) consensus in the field for the problem, or (3) direct citation of evidence of the problem (or some combination of all 3). I don’t think citing a single person’s opinion is good justification in general or intuitive to a reader presented with a new problem.
- Section 1.1: How does SKILL-MIX directly address the desiderata presented in the prior paragraph? For reader clarity, this question should be answered immediately after presenting the desiderata or else we will forget.
- Section 1.1: Arora & Goyal is cited as demonstrating that scaling up language models → improvements at applying k’-tuples of basic skills, therefore motivating the authors’ Skill-Mix benchmark. But this doesn’t mean that applying k’-tuples of basic skills → better general purpose understanding. This is a problem with many LLM benchmarks, but still presents an issue that the authors should discuss in this paragraph.
- Prior work: This section isn’t that well written. The authors should relate Skill-Mix to each of the types of prior work rather than describe them in isolation.
- Section 4: This is a minor issue, but Section 4 can probably be made a subsection of Section 3 as it is part of the introduced SkillMix evaluation.
- Table 1/2: Some sort of bolding on the best performing across each category for each k would be really helpful to parse all of these numbers
- Section 6: Can the authors explain the derivation of the high probability statement for $O(T \log T + N \log N)$? Intuitively, it’s quite surprising given “random prompts” to expose these low-probability skills and their topics.

**Questions:**

See weaknesses section

---

> ### Author Response · Authors · 2023-11-20
>
> **We thank the reviewer for the helpful and insightful comments! Please see our general response above and our responses to the reviewer’s main concern below:**
>
>
> > Skill picking: Section 3.1 (and the appendix) describe how the authors pick the skills by hand based on textbooks on logical reasoning, rhetoric, and theory of mind. But what is the justification for using these types of skills? For example, i’m sure there are many skills not related to any of those topics which still require some sort of intelligent capability to combine meaningfully. The authors should explain this better.
>
> We apologize for any confusion. The evaluation tests skills that are important in intelligence but is not meant to be a complete test of intelligence. Please see the response to all reviewers Q3.
>
>
> In Appendix C.2, we discuss details of our skill selection process and how we remove skills from the original large set due to various reasons. Conceptually, the choice of topic should not affect the use of language skills. The capability to combine seemingly unrelated skills and topics is indeed what Skill-Mix aims to test.

---

> > ### Author Response · Authors · 2023-11-20
> >
> > **Below, we address the other concerns brought up.**
> >
> > > One minor issue is that I don’t think the models picked necessarily need to be instruction-finetuned as regular, non-instruction finetuned language models are still able to follow instructions in the prompt. In fact, given that some recent work has demonstrated that instruction-finetuning on smaller models reduces their ability to produce as diverse of a response distribution, it could artificially hurt the performance of the selected open-source models in evaluation due to the selected topics occurring with a Google n-gram probability of ~1e-6 and them likely being fine-tuned on a much smaller dataset that may not include the selected topics/skills.
> >
> > We agree that the instruction following can work reasonably for many tasks in the base models. But Skill-Mix is  more complicated. Despite multiple attempts we were unable to get any reasonable performance on non-instruction tuned models, especially small 7B models. We would welcome any suggestions (e.g., prompting strategies)!
> >
> > > What is the variance among the automatically graded scores? It is stated that human variance is higher than the automatic ones, but neither seem to be presented anywhere.
> >
> > The standard deviation of GPT-4 grading is 0.050 (averaged across all points), which is much smaller than 0.261, the standard deviation of human grading as reported in Appendix B. The standard deviation of LLaMA-2 grading is 0.105, meaning it is noisier than GPT-4, but still much more stable than human grading.
> >
> > > The analysis of Skill-Mix vs other benchmarks could be more comprehensive — it would be interesting to see correlation scores against other benchmarks to see how skill-mix differs overall wrt individual LLM rankings.
> >
> > We agree with the reviewers that this would be ideal. However, none of the benchmarks have included all the models we tested, making it hard to compute the correlation. We find the ordering of the GPT family and the LLaMA family on Skill-Mix to be consistent with the ordering on other benchmarks. For ordering including newer models, we have pointed out all noticeable inversions in the paper.
> >
> >
> > > Writing + Clarity + Motivation:
> >
> > We thank the reviewer for their constructive comments, and have made appropriate changes in the revision (e.g., bolding on the best performing model; for this point in particular, we also include figures in the appendix to help visually assess how well models perform against each other for each metric).
> >
> > We apologize for the confusion with respect to the motivation, please see our general response, **Q3**.
> >
> > > Can the authors explain the derivation of the high probability statement for O(TlogT + NlogN) ? Intuitively, it’s quite surprising given “random prompts” to expose these low-probability skills and their topics.
> >
> > We apologize for the confusion. Here we are referring to a dataset that follows Skill-Mix, that is, the random prompts are sampled by uniformly picking $k$ skills out of $N$ total skills, and 1 topic out of $T$ total topics. Thus, each prompt reveals $k$ random skills and 1 random topic. By basic probability calculations, $O(T\log T+N\log N)$ random prompts should reveal all $N$ skills and $T$ topics with high probability even when $k=2$.
> >
> > > Intro: Why is a specific opinion from Hinton used as motivation for desiderata for the authors’ benchmark?
> >
> > We apologize for the confusion and have revised wording to following: “(d) capable of producing novel text generations corresponding to a mix of skills and a topic that it did not see in the training corpus.”  The ``stochastic parrots’’ question  (Bender et al., 2021)” has of course been debated for a few years. We cited the more recent (Hinton & Ng’23) conversation because Hinton emphasized that disagreements about this issue among experts is hindering a full discussion of AI safety. We would be happy to cite other sources if the reviewer has suggestions!
> >
> > > Tables: some sort of bolding on the best performing across each category for each k would be really helpful
> >
> > We thank the reviewer for pointing this out, and in our revision, we have included bolded numbers in Tables wherever appropriate!

---

### Official Review · Reviewer_NFpe · 2023-10-31

**Soundness:** 4 excellent
**Presentation:** 4 excellent
**Contribution:** 3 good
**Rating:** 8
**Confidence:** 4

**Summary:**

The paper presents a new evaluation benchmark SKILL-MIX for general-purpose language capabilities, specifically, a particular sort of compositional generalization. It tests the model’s ability to create text on-demand on a given topic and with a given combination of well-known skills. The paper provides an exemplary introduction and motivation for the task of appropriate evaluation of the newly emerging topics.  A key idea is that the skills and the topic are chosen randomly from a big list, which itself can be expanded in many ways to give new evaluations. Such evaluations may end up requiring the (Student) model to imagine and describe situations that does not exactly match anything seen during training. Grading this evaluation is shown to be possible with GPT-4 (and the authors spot-checked the grading). The authors also emphasized the important of human spot-checking, but since human evaluations were  found to have significant variance, they are not reported.

**Strengths:**

I found this paper one of the best papers I reviewed recently (including the Neurips reviewing).
I found the Introduction (Section 1) and especially "Desiderata for next-generation evaluations" extremely valuable for any LLM development. I am currently working on a task that requires complex evaluation of LLMs and the guidance and observations summarized in the introduction are right to the point of what we are looking for.

I also find the idea of mixing random skills with evolving N as a way to mitigate some of the issues in current evaluations simply, yet elegant.  The authors also validate its effectiveness with numerous experiments.

**Weaknesses:**

1) I'd love to see some correlation of the presented results with human spot-checking and evaluation and if such correlation does not exist, discussion on what might be the reason

2) I'd love to get some suggestion on how this framework can be extended beyond language to Vision - Language Models.

3) I'm curious to see how the experiments with fine-tuning while releasing 10% of skills pan out.

**Questions:**

1)Do you think that instead of have one strong evaluator, you can have a mixture of experts each capable to evaluate a single task exceptionally well?
2) Can you please provide reference for "While both models are creditable graders, they, like all current LLMs, were unreliable at simple arithmetic, which is important for calculating a total score."

---

> ### Author Response · Authors · 2023-11-20
>
> **We thank the reviewer for the helpful and insightful comments! Please see our general response above and our responses to the reviewer’s main concerns below:**
>
> > I'd love to get some suggestion on how this framework can be extended beyond language to Vision - Language Models.
>
> We sketch an example of how to extend Skill-Mix to evaluate vision-language models below:
>
> - The set of skills can include both vision skills (e.g., object recognition, object generation, 3D reconstruction, spatial relation between objects) and language/reasoning skills (e.g., description of the scene, object interaction, reasoning of past and future events).
> - The set of topics can be the environment or background of the image.
> - Task can either be answering questions based on given images, or generating images based on instructions (the instruction may include images as well).
> - The images fed to models (as part of the prompt) can be generated by graphics programs.
>
> > I'm curious to see how the experiments with fine-tuning while releasing 10% of skills pan out.
>
> We include some preliminary results on finetuning LLaMA2-7B on 50% skills and topics, and observe improvements across the various Skill-Mix metrics. We are excited to release a more detailed exploration of this in future work! Please also see our response to all reviewers, Q2.
>
> > Can you please provide reference for "While both models are creditable graders, they, like all current LLMs, were unreliable at simple arithmetic, which is important for calculating a total score”
>
> We provide some references below. We also include our experience in Appendix C.3.3.
>
> [1] Exposing Attention Glitches with Flip-Flop Language Modeling
> Bingbin Li, Jordan T. Ash, Surbhi Goel, Akshay Krishnamurthy, Cyril Zhang
>
> [2] Table 10, Chain-of-Thought Prompting Elicits Reasoning in Large Language Models
> Jason Wei, Xuezhi Wang, Dale Schuurmans, Maarten Bosma, Brian Ichter, Fei Xia, Ed H. Chi, Quoc V. Le, Denny Zhou
>
>
> ---
>
> **Below, we address the other concerns brought up.**
>
>
> > I'd love to see some correlation of the presented results with human spot-checking and evaluation and if such correlation does not exist, discussion on what might be the reason.
>
> We thank the reviewer for bringing up this point. In Appendix B, we mentioned that the difference between GPT-4 grading and the average human grading is ~0.25, meaning that the human and GPT-4 agree on most of the points. We will include details in the final version.
>
> > Do you think that instead of having one strong evaluator, you can have a mixture of experts each capable to evaluate a single task exceptionally well?
>
>
> This is a good point! If one decides to fine-tune the grader, it is unclear to us whether it is better to fine-tune one model or use a mixture of fine-tuned models. One concern of using a mixture of experts is that some experts may be harsher than others. However, in terms of human grading, we believe it would be more efficient to split skills and assign them to specific graders. We leave this a future work to explore.

---

### Official Review · Reviewer_YRGK · 2023-11-06

**Soundness:** 2 fair
**Presentation:** 3 good
**Contribution:** 2 fair
**Rating:** 6
**Confidence:** 3

**Summary:**

This paper presents a new general-purpose evaluation framework for LLMs called 'skill mix'. The approach is based on an N-choose-k sampling of a set of pre-defined 'skills', combined with a set of pre-defined topics. The authors present the prompting and evaluation configuration, an LLM-based automated evaluation methodology, empirical results of running this new benchmark against a collection of contemporary LLMs that score well on current leaderboards, and a limited set of human evaluations.

The language of the paper is reasonably clear, and lots of empirical evaluations have been performed, but I find the motivation and justification for the proposed framework lacking. The paper is a little under-prepared and would benefit from further work before publication.

**Strengths:**

* LLM evaluation (beyond statistical language modelling to general purpose planning systems) is an important and timely topic
 * In the empirical evaluation the authors have clearly made an effort to compare a wide range of contemporary LLMs that currently score well on leaderboards
 * The key concept behind the skill-mix framework is easy to understand

**Weaknesses:**

* The motivation for this evaluation framework needs further development. Sec 1. highlights accelerating rates of leaderboard saturation, training set contamination, and training corpora secrecy as pressing issues, and suggests 7 desiderata for new LLM evaluation frameworks; (a) relevant to general-purpose intelligence, (b) easy for humans to design and administer, (c) resistant to training-set contamination, (d) capable of revealing novelty in some sense, (e) easy to grade at scale, (f) easy to scale difficulty over time, and (g) comprehensible by the general public. This list seems promising, but the paper doesn't make it clear how skill-mix addresses each of these needs, or how these 7 desiderata are related to the 3 pressing issues with current evaluation methods. In particular, I see no reason to believe that, once released, skill-mix won't also suffer the problem of 'cramming for the leaderboard' (also see next point)
 * In the current form, skill-mix defends against 'cramming' by proposing a held-out private test set of skills and topics. However, any evaluation design that relies on private evaluation data is fundamentally flawed - the field of LLM evaluation design needs to learn from cryptography algorithm design on this count. This is also problematic from an open science perspective - making it harder for peer review to evaluate the efficacy of the evaluation framework for instance.
 * The theoretical justification for skill-mix is also unclear to me.
    * For instance - does it make sense to combine /any/ $k$ of the N skills together, or are there some incompatible skill-tuples?
    * The connections with pedagogy theory and the empirical observations from the Arora & Goyal 2023 paper are a good starting point, but it's not clear in what way combining skills relates to generalized intelligence. A clearer definition of 'skill' may help here.
 * Some 'late breaking' paper results are missing from the submitted version (e.g. footnote 3, Sec. 6) - the paper will be strengthened with the inclusion of these results in a future version.
 * (End of Sec. 5) "Difference between model ranking on popular LLM leaderboards vs. SKILL-MIX provide evidence of "cramming for the leaderboard", further validating that SKILL-MIX is a good evaluation benchmark" - the conclusion here doesn't follow from the premise.

**Questions:**

# Questions and comments

 * It is great to see some human experiments done to compare against the automated grading scheme (however I note the lack of IRB or ethics approval in the paper related to human experiments). However, I note that variance in human grading responses (e.g. see end of Sec. 4) shouldn't automatically be interpreted as lower quality of responses - the variance can often be a signal, not noise. For instance, this could indicate combinations of skills and/or topics which are perhaps less compatible or sensible to query LLMs with.
 * I like the core idea of N-choose-k as an evaluation approach that scales quickly with k. The paper and results gave me the impression that you only consider linguistic skills, which seems like a big limitation. I note though that in the appendix the list of 10 published skills includes several non-linguistic skills (e.g. folks physics etc.). It would make review and evaluation of skill-mix much clearer if the full list of skills were published, along with (more) examples of select skill-topic-tuples. In the current form, the limited (linguistic) skill-tuple examples make it hard to imagine what the range and scope of questions looks like.
 * A key part of the skill-mix framework needs to be how to do evaluation at scale, including human oversight. To this end, the paper should include a proposed strategy or curriculum for how the human 'spot checking' can be performed. Currently there is not detail about this (very important) aspect of the evaluation procedure.
 * Lots of interesting and important open questions are raised in the paragraph 'difficulty of grading' at the end of Sec. 6. In particular, it seems worth investigating the duality between grading vs. answering questions more generally as part of a theory of LLM pedagogy.
 * Additional experiments exploring what skill-mix would look like in a multi-modal setting are welcome in future work - this seems like a great area to explore next, as the authors note (end of Sec 7).

# Minor comments and grammatical points

 * Figure 1 caption - typo 'red hearing'.

**Details Of Ethics Concerns:**

This research included human subjects (PhD students and post-docs, see Sec. 4 and App. B), however is lacking any IRB or ethics approval details in the paper for this aspect of the experimental procedures. I am aware that research norms vary by geography, but from my perspective and in my country, this would count as a clear case of academic research misconduct requiring institutional review and/or disciplinary action from the host university. I defer to the ICLR ACs to determine if this constitutes a relevant breach of the conference code or not.

---

> ### Author Response · Authors · 2023-11-20
>
> We thank the reviewer for the helpful and insightful comments! Please see our general response above and our responses to the reviewer’s main concerns below:
>
> > lack of IRB or ethics approval
>
> We thank the reviewer for bringing up this point. The human graders were carrying out research too and we understand IRB is not required in such cases. The final version will fully explain this status.
>
> > the paper doesn't make it clear  … how these 7 desiderata are related to the 3 pressing issues with current evaluation methods.
>
> Please see our response to all reviews, **Q1**.
>
> > no reason to believe that, once released, skill-mix won't also suffer the problem of 'cramming for the leaderboard'
>
> We release 10 skills and 10 topics (randomly sampled)  to avoid “cramming” for Skill-Mix. For more detailed explanations, please see our response to all reviewers, **Q2**.
>
> > The theoretical justification for skill-mix is also unclear to me. For instance - does it make sense to combine any k of the N skills together, or are there some incompatible skill-tuples? The connections with pedagogy theory and the empirical observations from the Arora & Goyal 2023 paper are a good starting point, but it's not clear in what way combining skills relates to generalized intelligence.
>
> We apologize for any confusion and would like to emphasize that as of now, the goal of our evaluation is to test general-purpose text generation capability, rather than generalized intelligence. Please also see our response to all reviewers, **Q3**.
>
> With respect to incompatible skill tuples: We  eliminated  skills that in our judgement inherently compose poorly with other skills (see Appendix C.2). It is possible that this intuitive weeding still left some pair of skills that don’t compose well. But this would equally affect the final performance number of all models, and not be expected to affect the ranking.
>
> > I like the core idea of N-choose-k as an evaluation approach that scales quickly with k. The paper and results gave me the impression that you only consider linguistic skills, which seems like a big limitation.
>
> For our first version of Skill-Mix, we acknowledge that we only use language-related skills. However, we note that Skill-Mix can be expanded to other types of skills (e.g., domain-specific skills related to coding, science, economics, math, law, etc).
>
> ---
> Below, we address the other concerns brought up.
>
> > Some 'late breaking' paper results are missing from the submitted version (e.g. footnote 3, Sec. 6) - the paper will be strengthened with the inclusion of these results in a future version.
>
> We thank the reviewers for pointing this out, and have updated these results in our revised version of the paper. In particular,
> - For k = 5 and k = 6, we show that GPT-4 is generating correct answers a good fraction of time, and a reasonable portion (say more than 1/3) of these correct answers contain combinations of skills and topics that have never jointly appeared in the training.
>     - We verify this claim by providing upperbounds on the average frequency of skills, the average frequency of topics, and the number of sentences in the training corpus (see Sec. 6 and Appendix D).
> - We include preliminary finetuning results for LLaMA2 7b in Appendix E.
>
> >  (End of Sec. 5) "Difference between model ranking on popular LLM leaderboards vs. SKILL-MIX provide evidence of "cramming for the leaderboard", further validating that SKILL-MIX is a good evaluation benchmark" - the conclusion here doesn't follow from the premise.
>
>
> We apologize for any confusion. We meant to convey that we believe Skill-Mix is a good evaluation benchmark because model rankings on Skill-Mix coincide with anecdotal human experience with these models (and that this is not the case with model rankings on e.g., Open LLM). Many derivatives of Llama2 models scored higher  than the Llama2 model on the Open LLM leaderboard but performed much worse on Skill-Mix, suggesting their general-purpose language skills were harmed along the way.

---

### Author Response · Authors · 2023-11-20
**General Response to All Reviewers**

We thank the reviewers for their thoughtful reviews, and our encouraged by the fact that many of the reviewers find our work to be novel and simple, yet effective.

We have made revisions to our paper according to the reviewers’ comments. Below we address the major concerns:

**Q1**: Why is Skill-Mix resistant to data contamination and “cramming for the leaderboard”? How can Skill-Mix reveal novelty (i.e., verify whether models are beyond “stochastic parrots”)?

**A**: The reason is that Skill-Mix is implicitly testing capability on $\binom{N}{k} T$ tasks, (N = # of skills, T = # of topics). For even reasonable $N, k$, the number of such tasks is larger than the corpus size, and even larger than the size of all text data in the world. Thus it is extremely unlikely that a random combination of skills and topics used at test time have occurred in the training data. In the paper, N is relatively modest, around 10^2,  but this could be easily increased. As models get stronger, k could be increased too.  We have included more details about how to estimate the occurrence of text containing random combinations of skills and topics in our revision (Appendix D).

**Q2**: What is the reason behind releasing only 10% of the skills/topics set? How does the fine-tuning affect the performance on Skill-Mix?

**A**: As mentioned, some analysis of emergence in [AG23] suggests that with standard pretraining techniques, the ability to combine k skills emerges for larger k as the model is scaled up.  This seems to check out in our evaluations: N =100, k=3 was already too hard for most small models.  The open question is whether there exists a better or direct way to acquire this capability.   Our experiment with $k=2$ (included in the revision, Appendix E) shows that LLaMA-2-7B-chat fine-tuned on less than 1000 good examples of k=2 (provided by GPT-4)  can already slightly outperform LLaMA-2-70B-chat on Skill-Mix. This provides a way to “cram” for Skill-Mix if the set of skills and topics is known. (But this added proficiency might harm general-purpose text abilities.) We propose only releasing 10% of the skills/topics so that model creators do not cram for the particular set of skills used by our dataset. Instead, they should have to target the capability to combine skills more broadly.

We have also added evaluations based on 10% of the skills/topics for several models (see Appendix C.5). The resulting metrics are very close to the ones based on the full sets of skills/topics. This means the community can accurately estimate a model’s performance on Skill-Mix only based on the released skills and topics (as long as the model does not “cram” for the released skills/topics).

**Q3**: Does good performance on Skill-Mix imply good general intelligence?

**A**: No we only suggest the reverse direction: general intelligence implies the ability to combine skills. So **inability** to pass skillmix implies deficiencies in the model’s “intelligence.” But we do suggest that if the Skill-Mix framework is expanded in the future (hopefully, by many groups across the world) by greatly expanding the skills set being tested (e.g., coding, math, science, etc.) then passing those future variants of Skill-Mix will get closer to capturing general intelligence.

**Major revisions:**
- In Sec. 6 and Appendix D, we have added an upper bound for the probability of random skill/topic combinations appearing jointly in a short text of the training corpus.
- In Appendix E, we have added experiments of fine-tuning LLaMA2-7B-chat on good GPT-4 generations on Skill-Mix with $k=2$.
- In Appendix C.5, we have added an evaluation only using the released skills/topics.
- In Appendix C.6, we have added ablation of more strict versions of Skill-Mix.


We once again thank the reviewers for their detailed comments, suggestions, and feedback!

---

### Meta-Review · Area_Chair_66R2 · 2023-12-15

**Metareview:**

The basic idea here is to evaluate LLMs with a mostly-secret set of "skills" that can be recombined. The recombination means that we get a very large set of potential tests, making it very impractical to train on the test set even if it was available. The paper is well reasoned and interesting throughout.

The results are very interesting and might be worth the price of admission, I mean acceptance, on their own. They constitute fairly strong evidence that we are seeing Goodhart's law in action, where the benchmarks have been using as optimization targets to overall detrimental effect. Not releasing all of the skills publicly is a somewhat controversial decision, but there's a good motivation.

**Justification For Why Not Higher Score:**

Not clear that the paper actually measures the LLM competencies we are most interested in.

**Justification For Why Not Lower Score:**

This is a novel paper, well written, and with interesting results.

---

### Decision · Program_Chairs · 2024-01-16

Accept (poster)